



# Source Attribution of Ozone and Precursors in the Northeast U.S. Using Multiple Photochemical Model Based Approaches (CMAQ v5.3.2 and CAMx v7.10)

Qian Shu[1], Sergey L. Napelenok[1], William T. Hutzell[1], Kirk R. Baker[1], Benjamin Murphy[1], Christian
Hogrefe[1], Barron H. Henderson[1]

[1]U.S. Environmental Protection Agency, Research Triangle Park, NC, 27711, USA.

Correspondence to: Sergey L Napelenok (sergey.napelenok@epa.gov)

**Abstract.**

The Integrated Source Apportionment Method (ISAM) has been revised in the Community Multiscale
Air Quality (CMAQ) model. This work updates ISAM to maximize its flexibility, particularly for ozone
($O_3$) modeling, by providing multiple attribution options, including products inheriting attribution fully
from nitrogen oxide reactants, fully from volatile organic compound (VOC) reactants, equally to all
reactants, or dynamically to $NO_x$ or VOC reactants based on the indicator gross production ratio of
hydrogen peroxide ($H_2O_2$) to nitric acid ($HNO_3$). This study's primary objective is to document these
ISAM updates and demonstrate their impacts on source apportionment results for $O_3$ and its precursors.
Additionally, the ISAM results are compared with the Ozone Source Apportionment Technology
(OSAT) in the Comprehensive Air-quality Model with Extensions (CAMx) and the brute force method
(BF). All comparisons are performed for a 4km horizontal grid resolution application over the northeast
U.S. for a selected two-day summer case study (August 9th and 10th, 2018). General similarities among
ISAM, OSAT, and BF results add credibility to the new ISAM algorithms. However, some
discrepancies in magnitude or relative proportions among tracked sources illustrate the distinct features
of each approach while others may be related to differences in model formulation of chemical and
physical processes. Despite these differences, OSAT and ISAM still provide useful apportionment data
by identifying the geographical and temporal contributions of $O_3$ and its precursors. Both OSAT and
ISAM attribute the majority of $O_3$ and $NO_x$ contributions to boundary, mobile, and biogenic sources,
whereas the top three contributors to VOCs are found to be biogenic, boundary, and area sources.



# 1 Introduction

Tropospheric $O_3$ is a critical air pollutant that endangers human health (WHO, 2013) and sensitive vegetation (Booker et al., 2009), and contributes to climate change (Jacob and Winner, 2009).

It is produced through non-linear photochemical reactions of carbon monoxide (CO), volatile organic compounds (VOC), and nitrogen oxides ($NO_x = NO + NO_2$) with sunlight (Atkinson, 2000). In the United States, the national average ambient $O_3$ concentration has decreased by 22% since 1990, owing to regulations such as the Clean Air Act (CAA) on $NO_x$ and VOC emissions (Simon et al., 2015). Long-term space observations have also confirmed the improvement in air quality (Duncan et al., 2013;

Lamsal et al., 2015). However, many major metropolitan areas continue to exceed the $O_3$ national ambient air quality standards (NAAQS) set by the US Environmental Protection Agency (US EPA). To continue to reduce $O_3$ levels, it is critical to develop effective emission control strategies as has been done for other pollutants (Lefohn et al., 1998; Reitze, 2004; Cooper et al., 2015). The effectiveness of any $O_3$ control strategy hinges on accurately quantifying the contributions of various precursor

emissions to $O_3$ formation.

Numerous techniques have been used to characterize and quantify the relationship between emission sources and ozone concentrations, including statistical methods, model sensitivity simulations, and model source apportionment approaches, each with its own set of advantages and disadvantages (Cohan and Napelenok, 2011). While some traditional receptor-based methods based on chemical mass

balance (CMB, Hidy and Friedlander, 1971), such as Effective Variance solution (EV, Watson et al., 1984) and Positive Matrix Factorization (PMF, Paatero and Tapper, 1994), produce insightful results when measurements are taken at a specific receptor, they are typically applied to speciated VOC and particulate matter (PM) and are also constrained by the relative sparsity of observations in space and time, rendering them unsuitable for regional and national $O_3$ precursor emission control strategies.

Alternatively, three-dimensional air quality models (AQM) allow for the quantification of $O_3$ source contributions at regular intervals over longer periods and wider spatial distributions. The most basic source apportionment (SA) technique in the context of an AQM is to conduct source sensitivity simulations using the brute force (BF) method, in which several simulations are conducted, each with one source eliminated or reduced. The differences in the output fields compared to the baseline



simulation are then attributed to the eliminated or reduced source (e.g., Marmur et al., 2005). BF has

some limitations when used to determine total source culpability of $O_3$ due to the pollutants' nonlinear

dependence on both relative and absolute VOC and $NO_x$ concentrations. For example, at some chemical

regimes where the ambient ratio of total VOC to total $NO_x$ concentrations is sufficiently low, removing

$NO_x$ may lead to an increase in $O_3$ concentrations, because $NO_x$ also acts as a titrant of $O_3$. In some

cases, where a source contributes a substantial portion of total $NO_x$ or VOC emissions, complete source

removal for the purposes of source apportionment calculation may also substantially alter the

underlying chemical regime for formation of secondary pollutants such as $O_3$. Further, to separate the

contributions and interactions of "n" sources, Stein and Alpert (1993) showed that BF would require

two to the power of the number of sources ($2^n$). This is quickly impractical leading to a subset of BF

simulations with unknown interactions. As a result, summarizing the $O_3$ change in response to multiple

brute force emission source simulations can make it difficult to interpret the cumulative effect of those

emissions on $O_3$ (Kwok et al., 2015).

Reactive tracer or tagged species SA methods for $O_3$ have also been incorporated in AQMs.

These tracers are usually additional species added to the AQM to track the contributions of pollutants

from specific source categories. They undergo the same atmospheric processes as the bulk chemical

species within the model (Kwok et al, 2015). As one example, OSAT within CAMx quantifies the

contributions of various emission sectors, source regions, as well as initial and lateral boundary

conditions, to simulated $O_3$ concentrations (Ramboll Environ, 2015). OSAT allocates instantaneous $O_3$

formation to either $NO_x$ or VOCs based on the ratio of hydrogen peroxide ($H_2O_2$) to nitric acid ($HNO_3$)

production (Dunker et al., 2002). $O_3$ formation is classified as being $NO_x$-limited or VOC-limited based

on the gross production of $H_2O_2$ ($PH_2O_2$) and $HNO_3$ ($PHNO_3$). When the ratio ($PH_2O_2/PHNO_3$) is

below 0.35, the formation is classified as $NO_x$-limited and VOC-limited otherwise (Sillman, 1996). If

the photochemical formation of $O_3$ ($PO_3$) occurs in a $NO_x$-limited regime, the $NO_x$ tracers are used to

attribute $PO_3$ proportionally to the emissions sources that contributed to the $NO_x$ concentrations.

Otherwise, VOC tracers are used to attribute $PO_3$ to the sources that contributed to the VOC

concentrations (Dunker et al., 2002; Kwok et al., 2015). The OSAT formulation was recently changed

(OSAT3) to track all forms of $NO_x$ to account for $NO_x$ recycling, which occurs when $NO_x$ is converted



to another form of $NO_x$ (e.g., peroxyacetyl nitrate (PAN) or $HNO_3$) and then converted back to $NO_x$.

OSAT has been used to support policy assessments (e.g., U.S. EPA, state government agencies, etc.,

Ramboll Environ, 2015, 2022) as well as for scientific research purposes (Li et al., 2012; Zhang et al.,

2017; Shu et al., 2020).

Additionally, the Integrated Source Apportionment Method (ISAM) within CMAQ has shown

promising results for $O_3$ tagging (Kwok et al., 2013, 2015). Recent ISAM experiments have quantified

the contribution of $O_3$ sources to air pollution in several major cities throughout the United States and

Europe (Kwok et al., 2015; Valverde et al., 2016a; Karamchandani et al., 2017; Butler et al., 2018; Pay

et al., 2019). The attribution of $O_3$ and precursors from specific sources estimated by ISAM

implemented in version 5.0 of CMAQ compared well with source-specific aircraft transect

measurements (Baker and Kelly, 2014; Baker and Woody, 2017). The ISAM algorithms have also been

updated several times following the original implementation in CMAQv5.0.2.

ISAM updates presented here substantially increase the flexibility to the user of the CMAQ

source apportionment model and are described in detail below. Further in the manuscript we apply the

changes to CMAQ-ISAM for a Northeastern U.S. $O_3$ air quality episode and compare the results to

CMAQ-BF and CAMx-OSAT. The manuscript is organized as follows: Section 2 documents the ISAM

updates in detail; Section 3 describes the methodology for this study, which includes the base modeling

configurations, simulation designs for source apportionment, tracked species classes, evaluation

methods, and case study development; Section 4 presents the findings, including model evaluation

results and comparisons of source apportionment for several species; Section 5 documents the running

speed comparisons between CMAQ-ISAM, CAMx-OSAT and CMAQ-BF; and finally, the findings and

their implications for future research are discussed in Section 6.

## 2 Updates in ISAM

The ISAM implementation in the version 5.0 release of CMAQ was based on Kwok et al. (2013)

and Kwok et al. (2015). That approach was updated in CMAQ version 5.3 to an attribution based on

integrated reaction rates and product yields (US EPA, 2021). The 5.3 version of CMAQ-ISAM (US

EPA, 2022) employs an apportionment scheme that assigns products of each chemical reaction to



sources based on reactant stoichiometry (US EPA, 2021).  For example, the isoprene peroxy radical
($ISO_2$) reacts with nitric oxide (NO) to produce several different stable and radical species as
represented in the CB6R3 chemical mechanism by the following equation (Equation 1).

$$ISO_2 + NO = 0.1*INTR + 0.9*NO_2 + 0.673*FORM + 0.9*ISPD + 0.818*HO_2 + 0.082*XO_2H + 0.082*RO_2$$
Eq. (1)

In addition to nitrogen dioxide ($NO_2$), the products include isoprene nitrate (INTR), formaldehyde
(FORM), hydroperoxy radical ($HO_2$), alkoxy radicals ($XO_2H$), peroxy radical ($RO_2$), and other isoprene
reaction products (ISPD).  $ISO_2$ itself is a direct product from oxidation of isoprene in the chemical
mechanism, which, in turn, originates from overwhelmingly biogenic sources. Conversely, NO is
typically emitted from anthropogenic combustion processes, with a much smaller natural component

originating from lightning strikes and microbial soil processes (Jacquemin et al., 1990; Yienger et al.,
1995, Pierce et al., 1999) on the global scale. Thus, the reactants of this reaction are approximately half
from biogenic and half from anthropogenic sources resulting in the products listed above inheriting the
same attribution distribution.  However, source attribution approaches, both receptor-based such as
PMF and source-based such as ISAM, are often used to understand how originally emitted $NO_x$ and

VOC from particular sources ultimately contribute to model predicted $O_3$ production. Therefore, the
loss of source identity through processes, such as the photo-stationary state (PSS) null cycle (Leighton,
1961) or enhancing the influence of sources that are not controlling $O_3$ production due to nonlinear
chemistry, could shift culpability of emissions sources. In the example above, $NO_2$ produced by
Equation 1 inherits the approximately 50% biogenic and 50% anthropogenic source assignment. These

types of source assignments can propagate quickly in situations where the PSS causes $NO_2$ to cycle
back to NO through photooxidation and radical oxidation with increasingly higher attribution to
biogenic sources. Since $NO_x$ cycling is a fast process in the context of regional air pollution modeling,
anthropogenically emitted nitrogen species can become assigned to biogenic (or other nearby) sources
downwind because the original emitted source identity was not retained through these complex

reactions. Equation 1 is just one example that illustrates the complex relationship between precursors
and subsequent source identities of secondary pollutants.  Many such reactions exist in modern
chemical mechanisms, which themselves are a simplification of the actual atmosphere. In some source





apportionment applications, such as in the context of $O_3$ source attribution assessments, nitrogen molecules should retain their original source signatures. This is also the approach by other

apportionment models such as OSAT, earlier ISAM implementations (Kwok et al, 2015) and other tagging methods such as those developed by Butler et al (2018) and Grewe et al (2010).

Since attribution objectives may vary based on scale (e.g., global compared to urban) or purpose (e.g., policy or tracing chemical reactions), ISAM has been enhanced to provide additional configuration options for the user to define how secondarily formed gaseous species are assigned to

sources of parent reactants (Table 1) (US EPA 2022). The original assignment scheme with equal source assignment based on reaction stoichiometry introduced in CMAQ version 5.3.2 (US EPA 2021) was retained as ISAM-OP1. ISAM-OP2 was added to give the user the option to always assign products to sources emitting nitrogen reactants including NO, $NO_2$, nitrate radical ($NO_3$), nitrous acid (HONO), and/or aerosol nitrate ($ANO_3$). Sources without these species utilize an assignment approach

consistent with ISAM-OP1. ISAM-OP3 was added to include the nitrogen species from ISAM-OP2 with the addition of all the reactive VOCs and intermediary organic radicals participating in ozone chemistry as initial source assignment criteria, completing the list of species important in $O_3$ chemistry. ISAM-OP4 was added to allow the user to assign product to only sources with the reactive VOCs and radicals if present, and with an ISAM-OP1 assignment if not present. Finally, ISAM-OP5 was added to

make the assignment decision based on the ratio of $PH_2O_2$ to $PHNO_3$ (Sillman, 1996). Assignment is made to sources with NOx species (ISAM-OP2) or to VOC species (ISAM-OP4) considering whether the $O_3$ producing chemical regime is $NO_x$-limited or VOC-limited with the $PH_2O_2/PHNO_3 = 0.35$ as the threshold. These CMAQ-ISAM options, including the regime threshold value (or transition point), are accessible to users at runtime through the standard model run script.


**Table 1. Expanded CMAQ-ISAM options.**

| CMAQ ISAM option | Description |
|---|---|
| ISAM-OP1 | Source attribution uniformly based on stoichiometric reaction rate products. |
| ISAM-OP2 | Assignment to sources with NO, $NO_2$, $NO_3$, HONO, or $ANO_3$ if present in parent reactants, otherwise proportional assignment as ISAM-OP1. |





| ISAM-OP3 | Assignment to sources with species from ISAM-OP2 in addition to reactive VOC species* and radicals if present in parent reactants, otherwise proportional assignment as ISAM-OP1. |
| ISAM-OP4 | Assignment to sources with reactive VOC species and radicals if present in parent reactants, otherwise proportional assignment as ISAM-OP1. |
| ISAM-OP5 | Assignment based on the ratio of $PH_2O_2$ to $PHNO_3$ for species in ISAM-OP3. |
| VOC-$NO_x$ limiting Transition Point | Value of the above ratio where assignment changes from species groups (default 0.35). |

*Detailed species assigned to sources for each option are listed in Table S3.

## 3 Method

### 3.1 Base model configurations

Two models, CMAQ version 5.3.2 with modified ISAM and CAMx version 7.10 with OSAT3, are used to simulate a one-month period during the summer of 2018 (July 29th to August 30st). Both models are applied to the same horizontal modeling domain with 4 km x 4 km resolution encompassing the northeastern United States. This domain is nested within a larger 12 km domain that encompasses the entire contiguous United States which is used for providing simulation boundary and initial

conditions (BC and IC) for the 4 km domain. BCs were generated for the 12 km simulations using a hemispheric application of the GEOS-Chem model (Henderson et al., 2014) that was run for 2018. Anthropogenic emissions were based on version 1 of the 2016 National Emission Inventory (NEI, US EPA, 2021). Electrical Generating Unit emissions were based on continuous emissions monitoring data from 2018 where available. Onroad emissions were projected to 2018 to reflect decreases in emissions

due to vehicle fleet turnover and the implementation of emission control technology in 2017. The Biogenic Emission Inventory System (Bash et al., 2016) was used to generate biogenic volatile organic compound emissions, and offline meteorology was created using the Weather Research and Forecasting (WRF, Skamarock et al., 2008) model version 3.8. CMAQ was configured using Carbon Bond 6 version 3 (CB6R3, Emery et al., 2015) for chemistry. Similarly, all base meteorological and emissions

inputs for CAMx were identical to those for CMAQ but were processed using CAMx appropriate data pre-processors (https://www.camx.com). The CAMx model was configured with Carbon Bond 6





version 4 (CB6R4, Emery et al., 2016) chemical mechanism. Table 1 contains the summary of the two model configurations.

**Table 2. CMAQ and CAMx model configurations**

| Model option | CMAQ | CAMx |
|---|---|---|
| Model version | Version 5.3.2 | Version 7.10 |
| Horizontal resolution | 4 km | 4 km |
| Vertical layers | 35 | 35 |
| Meteorology | WRF3.8 | WRF3.8 |
| Anthropogenic Emissions | 2016 NEI version 1[a] | 2016 NEI version 1[b] |
| Biogenic Emissions | BEIS[c] | BEIS[c] |
| BC/IC | 12km US CONUS | 12km US CONUS |
| Gas phase chemistry | CB6R3 | CB6R4 |
| Source apportionment | ISAM | OSAT3 |

[a]EGU were based on continuous emissions monitoring data from 2018 where available. Onroad emissions were projected to 2018.
[b]CAMx EGU and Onroad were identically processed as CMAQ.
[c]BELD v4.1 vegetation data for biogenic emissions, BEIS version is 3.61.

## 3.2 Source apportionment simulation designs

190       As discussed in Section 2, ISAM has been updated to include a user option with five possible configurations for source apportionment approach. Here, we conduct CMAQ source apportionment simulations for all these options: ISAM-OP1, ISAM-OP2, ISAM-OP3, ISAM-OP4 and ISAM-OP5, hereafter referred to as OP1, OP2, OP3, OP4 and OP5.

      Additionally, the OSAT3 approach was used in the CAMx v7.10 base model for comparison

with the five ISAM simulations. Hereafter OSAT3 is referred to as OSAT. For better understanding the differences between ISAM options and OSAT used in this analysis, the source apportionment approach implemented in CAMx is briefly recapped here. All available versions of OSAT (including OSAT3) in CAMx separately solve for production and destruction of $O_3$ with production being attributed to either $NO_x$ or VOC emissions, depending on which is estimated to be limiting $O_3$ production. When the ratio

of $PH_2O_2/PHNO_3$ exceeds 0.35, the produced $O_3$ is attributed to VOC emissions, and $NO_x$ emissions below that threshold. The CAMx source apportionment implementation includes an option (OSAT-



APCA) that allows for a redirection of attribution to anthropogenic emissions in situations where the limiting precursor is biogenic. That option was not used for this analysis.

In OSAT, $O_3$ attributed to $NO_x$ and VOCs is tracked as separate tracer groups. $O_3$ tracers are

first adjusted to account for $O_3$ destruction processes and subsequently for net $O_3$ production, which is defined as the difference between $O_3$ production and $O_3$ destruction based on a subset of photochemical reactions that result in $O_3$ destruction. In situations where the net $O_3$ production is negative (destruction reactions dominate), all the $O_3$ tracers are proportionally decreased. When net $O_3$ production is positive, production is assigned proportionally to the sources of those emissions ($NO_x$ and VOC precursor

tracers) at the time and place where $O_3$ was made. OSAT includes a group of tracers that track odd-oxygen that is consumed when $O_3$ reacts with NO to form $NO_2$ that can quickly photolyze and reform $O_3$ through a reaction with oxygen. In this situation, the $O_3$ removed from the $O_3$ tracers due to the NO + $O_3$ reaction is moved to the odd-oxygen tracers (which have separate $NO_x$ and VOC tracer groups). When $NO_2$ is photolyzed and $O_3$ formed a proportional amount of $O_3$ is taken from the odd-oxygen

tracers and moved to the $O_3$ tracers.

Finally, a brute force method (zeroing out the entire emission stream for tracked sources in CMAQ, hereafter referred to as CMAQ-BF) was also used to compare with the ISAM options and OSAT. Eleven different emission source categories were tracked using each apportionment technique. The source categories comprise four point-source categories including electricity generating units

(EGU), non-electricity generating units (NONEGU), fires (FIRE), and commercial marine vessels (CMV), and six area-source categories including on-road mobile (ONROAD), non-road mobile (NONROAD), biogenic (BIO), railway (RAIL), airports (AIRP), and other (AREA). Additionally, OILGAS was tracked as a mixed category (both point and area) of emissions from the oil and natural gas industry in the domain. Total emissions from the above sectors have been displayed in Table 3.

Finally, three predefined tracers for lateral boundary conditions (BCON), initial conditions (ICON), and other sources (OTHR) were also tracked for $O_3$ and its precursors. OTHR is used for all remaining untagged emission categories. For CMAQ-BF, a unique CMAQ simulation for each emission source category listed above was performed by fully removing the category's entire emission stream. CMAQ-BF apportionment was then calculated by subtracting the resulting pollutant fields from a base model



simulation. However, for ICON and BCON, each was reduced by 50%, and the output field difference

with the base model was scaled up by a factor of 2 to avoid numerical issues associated with very low

model ICON and BCON values. As for OTHR, there is no suitable way to retain an appropriate

chemical state of the troposphere after subtracting necessary emission categories, initial and boundary

conditions from an original CMAQ simulation. Thus, OTHR is not being compared among CMAQ-BF,

ISAM and OSAT in this study.

**Table 3. Total emissions from each sector for 4km Northeast U.S. domain (month of August 2018)**

|  | Tons/month | | Percent of Total (%) | |
|---|---|---|---|---|
| Sector | NOx | VOC | NOx | VOC |
| AIRP | 2,536 | 1,198 | 1.6 | 0.1 |
| AREA | 10,617 | 95,434 | 6.8 | 8.7 |
| BIO | 8,721 | 895,829 | 5.5 | 81.6 |
| CMV | 6,262 | 684 | 4.0 | 0.1 |
| EGU | 22,458 | 791 | 14.3 | 0.1 |
| FIRE | 400 | 5,007 | 0.3 | 0.5 |
| NONEGU | 15,020 | 11,323 | 9.6 | 1.0 |
| NONROAD | 23,958 | 33,561 | 15.2 | 3.1 |
| OILGAS | 11,053 | 22,526 | 7.0 | 2.1 |
| ONROAD | 49,361 | 30,578 | 31.4 | 2.8 |
| RAIL | 6,847 | 318 | 4.4 | 0.0 |
| Total | 157,233 | 1,097,247 | 100 | 100 |

## 3.3 Tracked species classes

O$_3$, NO$_x$ and VOC species were tracked by each method. As mentioned above, ISAM tracks all

individual oxidized nitrogen and VOC species, whereas OSAT tracks tracer families for each. To

facilitate the comparison between the two models, the ISAM species were aggregated in the same

fashion as OSAT (Table 4). However, some differences still exist since the two models have distinct

species representation. The nitrogen groupings NO$_y$ and RNO$_x$ (Table 4) were added to better elucidate

the behavior of each model under different O$_3$ producing chemical regimes.

**Table 4. Tracked species classes between ISAM and OSAT.**





| OSAT | ISAM |
|---|---|
| $O_3$ | $O_3$ |
| $RGN=NO_2+NO_3+2*N_2O_5+INO_3$ | [1]$RGN=NO_2+NO_3+2*N_2O_5$ |
| $NIT=NO+HONO$ | $NIT=NO+HONO$ |
| $TPN=PAN+PNA+PANX+OPAN+INTR$ | [2]$TPN=PAN+PNA+PANX+INTR$ |
| $NTR=NTR_1+NTR_2+CRON$ | [3]$NTR=NTR_1+NTR_2$ |
| $HNO_3$ | $HNO_3$ |
| $RNOx=RGN+NIT$ | $RNOx=RGN+NIT$ |
| $NOy=RGN+NIT+TPN+NTR+HNO_3$ | $NOy=RGN+NIT+TPN+NTR+HNO_3$ |
| [4]$VOC=1.0*PAR+1.0*MEOH+1.0*FORM+1.0*KET+2.0*ETHA$ $+2.0*ETOH+2.0*ETH+2.0*OLE+2.0*ALD_2+2.0*ALDX+2.0*E$ $THY+3.0*PRPA+3.0*ACET+4.0*IOLE+5.0*ISOP+6.0*BENZ+$ $7.0*TOL+8.0*XYL+10.0*TERP$ | $VOC=1.0*PAR+1.0*MEOH+1.0*FORM+1.0*KET+2.0*ETH$ $A+2.0*ETOH+2.0*ETH+2.0*OLE+2.0*ALD_2+2.0*ALDX+2.$ $0*ETHY+3.0*PRPA+3.0*ACET+4.0*IOLE+5.0*ISOP+6.0*B$ $ENZ+7.0*TOL+8.0*XYL+10.0*TERP$ |

[1]ISAM does not track $INO_3$
[2]ISAM does not track OPAN
[3]ISAM does not track CRON
[4]OSAT VOC has been pre-calculated as equation in Table 4

## 3.4 Evaluation method and case study development

Although identical emissions and meteorological inputs are used for CAMx and CMAQ (Table 2), potential differences still exist in multiple scales and processes. Shu et al. (2017, 2022) have reported that deposition is one of the largest uncertainties between the two models when other processes

are constrained. For inter-comparing ISAM and OSAT, it is not feasible to constrain all process uncertainties. Thus, we established criteria to choose representative days for ISAM and OSAT comparisons based on the performance of their parent models rather than comparing them throughout the entire simulation period to reduce the difference that may be brought on from their parent models. We initially set the correlation relationship ($R^2$) criteria between CMAQ and CAMx to be above 0.7 to

ensure that the performance of the two parent models is comparable. Next, maximum daily 8-hour averaged (MDA8) $O_3$ was used as the indicator for case study selection since ISAM and OSAT normally are used as regulatory application with this metric. We assess the mean bias (MB) of MDA8 $O_3$ for every day to choose the days on which both models have the lowest MB for predicted MDA8 $O_3$. Therefore, CMAQ and CAMx simulated ambient concentrations were paired in space and time with

observed data from the Air Quality System (AQS, https://www.epa.gov/aqs) monitoring network.



Hourly concentrations of total $O_3$, NO and $NO_2$ were also compared to the AQS observations, and their bias statistical metrics were calculated as well.

## 4 Results

### 4.1 Model performance evaluation and case study selection

270       Figure 1 shows observed site averaged MDA8 $O_3$ and its corresponding biases predicted by CMAQ and CAMx over paired AQS sites for the entire episode. Observed site averaged MDA8 $O_3$ ranges from 30 to 50 ppbv. The performance of two models for predicting MDA8 $O_3$ varies by paired day and monitor site with the range of biases from -23 to 35 ppbv, approximately. Table S1 summarizes $R^2$ and MB of MDA8 $O_3$ for each day for both models. Based on our criteria introduced in Section 3.4,

there are 13 days on which the two models show very good correlation relationships. Among these days, two models both show good performance on predicting MDA8 $O_3$ with closest MB on Aug 09[th] (CMAQ/CAMx = 3.09/2.99 ppbv) and 10[th] (CMAQ/CAMx: 2.42/2.61 ppbv). For other days, either two models both have higher MB (> 10 ppbv) or they have inconsistent predicted concentrations. Therefore, Aug 09[th] and 10[th] were selected as a two-day case study for source apportionment comparisons.

Additional evaluations of hourly $O_3$, NO and $NO_2$ is available in Fig. S1 of the supplemental information (SI). From Fig. 2, MDA8 $O_3$ is relatively higher over east coastal urban areas with generally over 50 ppbv but reduces to 35 ppbv at other rural areas of northeast U.S. domain. The two models predicted MDA8 $O_3$ show very good agreement spatially, underestimating MDA8 $O_3$ at sites where observed MDA8 $O_3$ is high but overestimating MDA8 $O_3$ at sites where $O_3$ is low. Similar spatial

plots of hourly paired $O_3$, NO and $NO_2$ can be found in SI (Fig S2). Table 5 and 6 respectively summarize statistical metrics for MDA8 $O_3$, hourly $O_3$, NO and $NO_2$ at all paired monitoring sites for the monthly $O_3$ episode and the selected two-day case study episode.

      The metrics in Table 5 and 6 both show consistent results with Fig. 1 as discussed above. The changes of NO and $NO_2$ metrics are marginal from the monthly episode to the two-day case. As in Fig

S1, NO and $NO_2$ concentrations are less variable than $O_3$ across days in the monthly episode, as a result, the comparison of NO and $NO_2$ are less dependent on which day is selected. Unlike NO and $NO_2$,





CAMx and CMAQ performance is statistically better in the two-day case study with lower MB for hourly $O_3$ (CMAQ/CAMx = 4.67/7.02 ppbv) and MDA8 $O_3$ (CMAQ/CAMx = 2.75/2.80) than the monthly episode (hourly $O_3$: CMAQ/CAMx = 6.49/7.99 ppbv; MDA8 $O_3$: CMAQ/CAMx = 5.30/4.18).

The differences of MB, NMB and $R^2$ between the two models also diminish from the monthly episode to the two-day episode. The statistical metrics of hourly O3 and MDA8 O3 demonstrate that the selected two-day case is suitable for a source apportionment comparison in which CAMx and CMAQ not only both have the least-biased predictions compared to observations but also show a good agreement with each other.

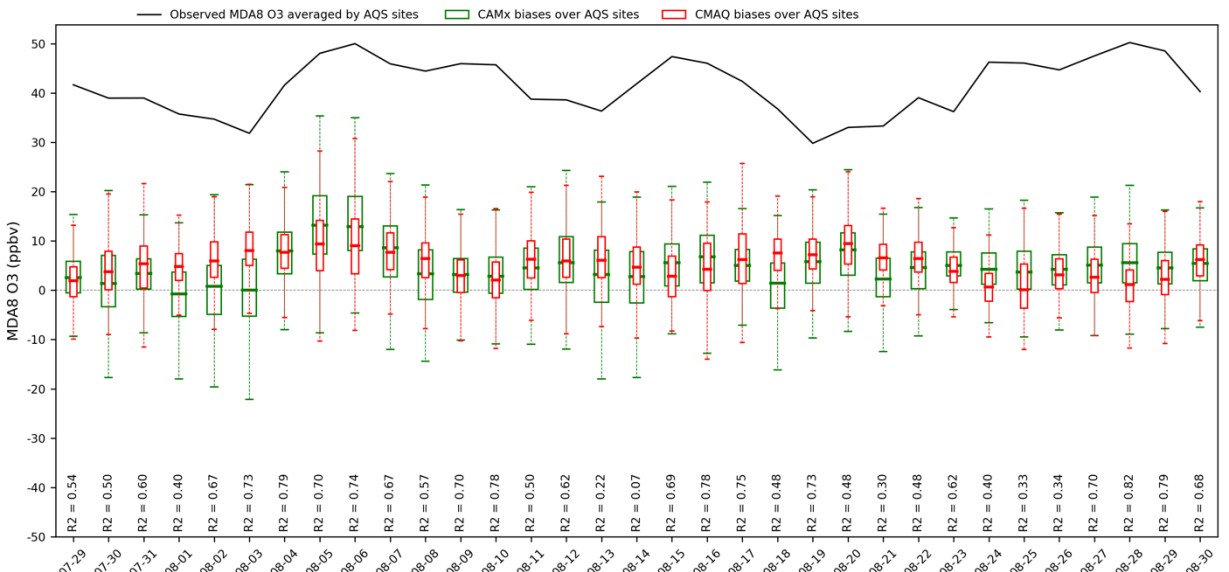


**Fig.1 observed site averaged MDA8 $O_3$ and its corresponding biases predicted by CMAQ and CAMx over paired AQS sites for the entire episode. $R^2$ shows correlation relationship between CMAQ and CAMx.**





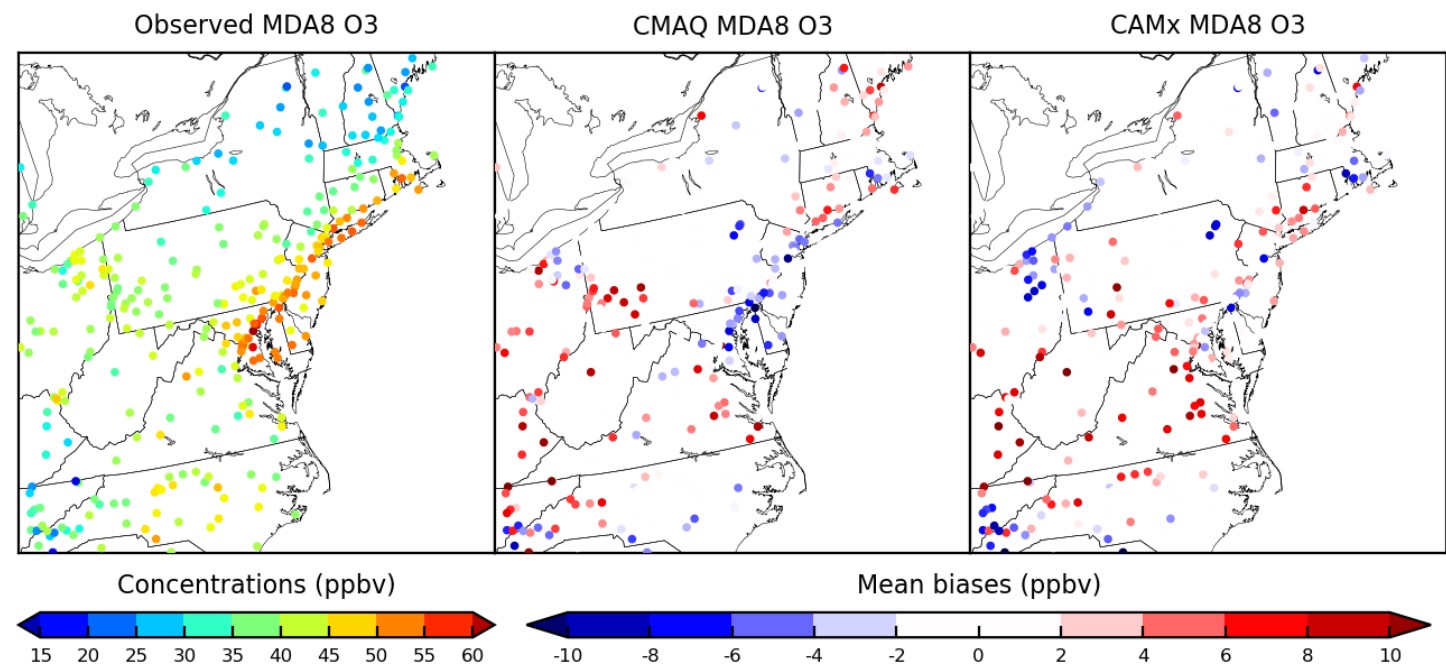


**Fig.2 Two-day averaged observed MDA8 O₃ over paired sites for northeast US domain and its corresponding mean biases predicted by CMAQ and CAMx for selected case study**

**Table 5. Model performance summary at paired AQS surface monitoring sites. (Monthly episode)**

| Species | Model | Number of Observations | MB[a] | NMB[b] | RMSE[c] | [d]$R^2$ |
|---|---|---|---|---|---|---|
| Hourly NO | CMAQ | 72987 | -1.05 | -44.50 | 6.24 | 0.07 |
| | CAMx | 72987 | -1.23 | -52.25 | 6.39 | 0.05 |
| Hourly NO₂ | CMAQ | 61987 | 0.64 | 10.21 | 6.39 | 0.32 |
| | CAMx | 61987 | 1.86 | 29.78 | 7.57 | 0.28 |
| Hourly O₃ | CMAQ | 232768 | 6.49 | 23.11 | 11.73 | 0.59 |
| | CAMx | 232768 | 7.99 | 28.47 | 14.46 | 0.42 |
| MDA8 O₃ | CMAQ | 9409 | 5.30 | 12.80 | 8.23 | 0.64 |
| | CAMx | 9409 | 4.18 | 10.09 | 9.26 | 0.58 |


**Table 6. Model performance summary at paired AQS surface monitoring sites. (Two-day case study episode)**

| Species | Model | Number of Observations | MB | NMB | RMSE | $R^2$ |
|---|---|---|---|---|---|---|
| Hourly NO | CMAQ | 4264 | -1.15 | -48.30 | 6.44 | 0.05 |
| | CAMx | 4264 | -1.38 | -58.14 | 6.57 | 0.04 |
| Hourly NO₂ | CMAQ | 3612 | 0.15 | 2.20 | 6.83 | 0.28 |
| | CAMx | 3612 | 0.83 | 11.88 | 7.51 | 0.25 |
| | CMAQ | 13486 | 4.67 | 15.06 | 10.88 | 0.61 |





| | | | | | | |
|---|---|---|---|---|---|---|
| Hourly O$_3$ | CAMx | 13486 | 7.02 | 22.65 | 13.26 | 0.49 |
| MDA8 O$_3$ | CMAQ | 567 | 2.75 | 6.00 | 6.28 | 0.62 |
| | CAMx | 567 | 2.80 | 6.10 | 6.95 | 0.63 |

[a] Mean bias: $MB = \frac{1}{N} \sum M_i - O_i$, MB ranges from negative infinity to positive infinity with 0 indicating unbiased data, unit here is ppbv.

[b] Normalized mean bias: $NMB = \frac{1}{N} \sum \frac{M_i - O_i}{O_i}$, ranges from negative 1 to positive infinity with 0 indicating unbiased data. The values shown in the table were multiplied by 100.

[c] Root mean square error: RMSE $= \sqrt{\frac{1}{n} \Sigma_{i=1}^{n}(M_i - O_i)^2}$, is the standard deviation of the prediction errors.

[d] $R^2 = \left\{ \frac{\sum(O_i - \bar{O})(M_i - \bar{M})}{\sqrt{\sum(O_i - \bar{O})^2 \sum(M_i - \bar{M})^2}} \right\}^2$, R$^2$ ranges from 0 to 1 with 1 indicating perfect correlation and 0 indicating an uncorrelated relationship.


## 4.2 Comparison of model source apportionment

### 4.2.1 Temporal variations of sector contributions

To better understand how the ISAM model apportionment approach simulated source contributions at each time step, time-series comparisons for each source were examined for O$_3$ and its

precursors, RNO$_x$ and VOC for the two-day case study. Figure 3 shows hourly variations of domain averaged predicted total O$_3$ (bulk) concentrations and sector contributions for seven source apportionment simulations (OSAT, BF, ISAM OP1 to OP5). In Fig. 3, CMAQ and CAMx predict similar ozone concentrations during the day, but differences appear at night, with a maximum difference of 5 ppb. This disparity was discussed in Section 4.1 and can be mitigated by employing the MDA8 O$_3$

metric. The seven source apportionment simulations yield similar diurnal trends via the trajectory of the total concentrations, but they apportion concentrations to each sector somewhat differently. Comparisons of five ISAM options reveals significant variability. OP1, which apportions uniformly according to stoichiometry, shows similar trends of apportionments for each sector as OP4, an option that always allocates products to sources with reactive VOCs and their radicals. They both apportion

more BCON and BIO O$_3$ but fewer contributions from all other sectors than the other three ISAM options (OP2, OP3 and OP5). Results of OP1 and OP4 would likely overestimate sensitivity to emissions to these reactants because VOCs are often available in excess. OP2 always allocates products to sources with nitrogen reactants, which prevents the attribution of NO$_x$ to non-nitrogen reactants.





Typically, these non-nitrogen reactants are common in transported (e.g., BCON) or natural sources
(e.g., isoprene in BIO). As a result, OP2 decreases BCON and BIO contributions while increasing
contributions from other sectors relative to OP1 and OP4.

OP5 assigns products to either reactive VOCs or $NO_x$ based on the ratio of $PH_2O_2/PHNO_3$,
placing $O_3$ contribution results for all sectors between the previous four ISAM options. OSAT, which
utilizes a similar methodology as OP5, shows consistent diurnal patterns of domain averaged total $O_3$
and sector contributions as the ISAM options, but with varying magnitudes. OSAT has the largest
BCON $O_3$ but the least contributions from AREA, BIO and FIRE. The rest of the OSAT sector
contributions are between the ISAM options. Consistent with earlier findings, CMAQ-BF estimates
systematically smaller $O_3$ contributions for all sectors besides EGU and BCON (Kwok et al., 2015).
While ISAM and OSAT appear to retain bulk mass as intended, CMAQ-BF shifts the chemical system
into a different, typically negative, nonlinear $O_3$ response to source change.

In Fig. 4, CAMx and CMAQ predict comparable total $RNO_x$ except for the first 12 hours of the
two-day example, when OSAT values deviate from those of the other six simulations. As the total
concentrations of the two models converge, OSAT exhibits similar patterns to OP2 and OP3. OP1, OP4
and OP5 show comparable results, with increased BCON and BIO $RNO_x$ but decreased contributions
from other sectors. Except for BCON and BIO, CMAQ-BF results are comparable to OP2 and OSAT
for all sectors. CMAQ-BF show comparable results with OSAT, OP2 and OP3 except for BCON and
BIO, which are negative for CMAQ-BF, suggesting that removing these source sectors results in a
slight rise in $RNO_x$. In previous source sensitivity and allocation investigations, it has been shown that
BF may have limits when the model response contains an indirect effect coming from the influence of
substances other than the direct precursors (Kwok et al., 2015; Burr and Zhang, 2011; Koo et al., 2009;
Jimenez and Baldasano, 2004; Zhang et al., 2009). This would be particularly true in situations where
emissions are a large percentage of total $NO_x$ or VOC in a particular area. The nonlinear impacts on gas
phase chemistry realized in a source sensitivity model simulation would not be a relevant representation
of culpability from that same source group.

Figure 5 illustrates the hourly variability of domain-averaged VOC concentrations and sector
contributions. CAMx only gives pre-lumped VOC (Table 4) for OSAT outputs. For consistency, VOC




for CMAQ ISAM and BF has also been carbon-weighted by summing all individual VOC species in CMAQ outputs using the same method as OSAT (Table 4). In Fig. 5, CAMx consistently simulates higher attribution to total VOC concentrations than CMAQ, with a maximum difference of 30 ppb.

These larger CAMx VOC concentrations are also reflected in apportioned OSAT sectors, particularly those with substantial contributions, such as BCON and BIO. The five ISAM options have comparable diurnal patterns for most sectors, with the exception of CMV, EGU, and RAIL, however the magnitudes for these three sectors are relatively minor, which is consistent with earlier findings (Kwok et al., 2015). CMAQ-BF estimates notably lower sector contributions for VOCs, which is similar to $O_3$ results (Fig.

4), with negative contributions for small sectors (e.g., CMV, EGU, and RAIL). Additional figures of other grouped nitrogen species tracked in Table 4 (e.g., RGN, $HNO_3$ and $NO_y$) can be found in SI.

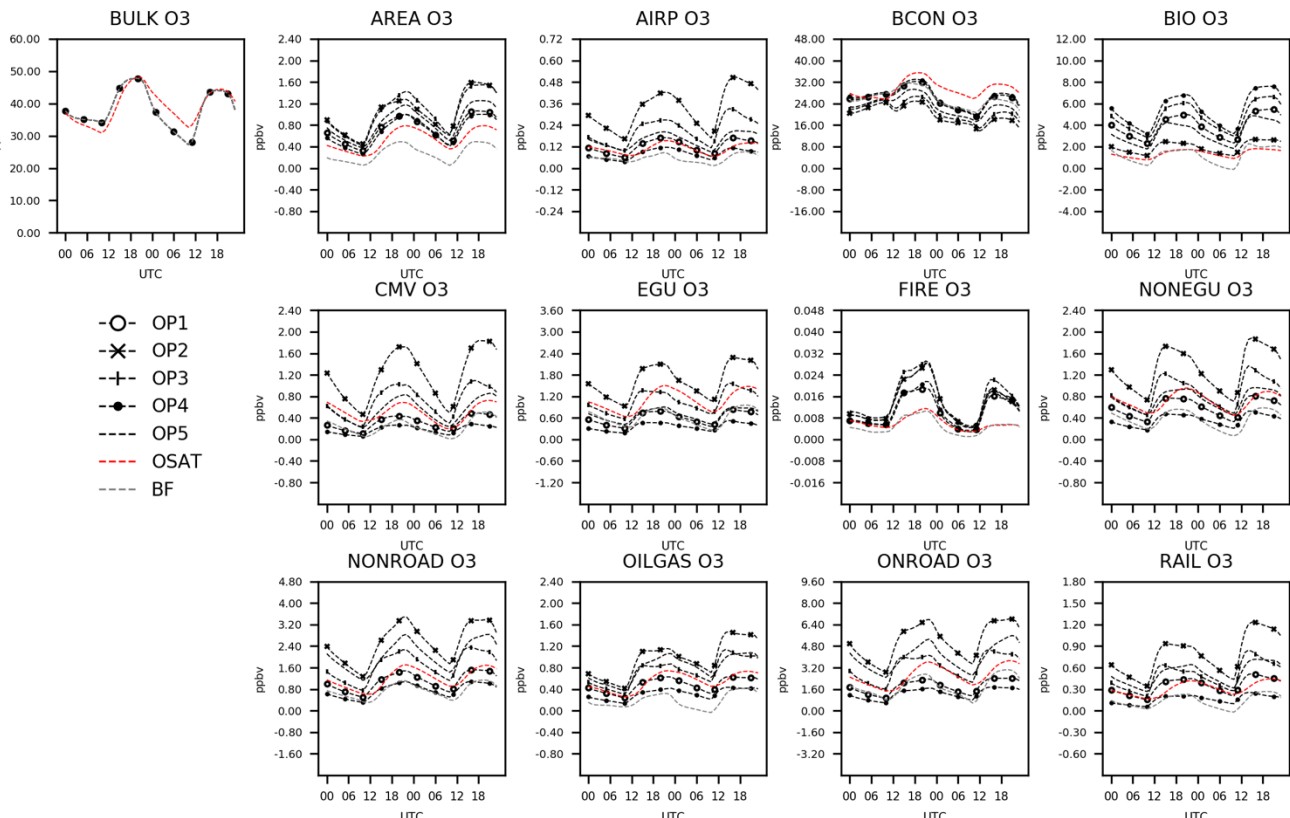

**Fig.3 Total and attributed $O_3$ concentrations to various sectors as a function of hour of day and apportionment technique.**



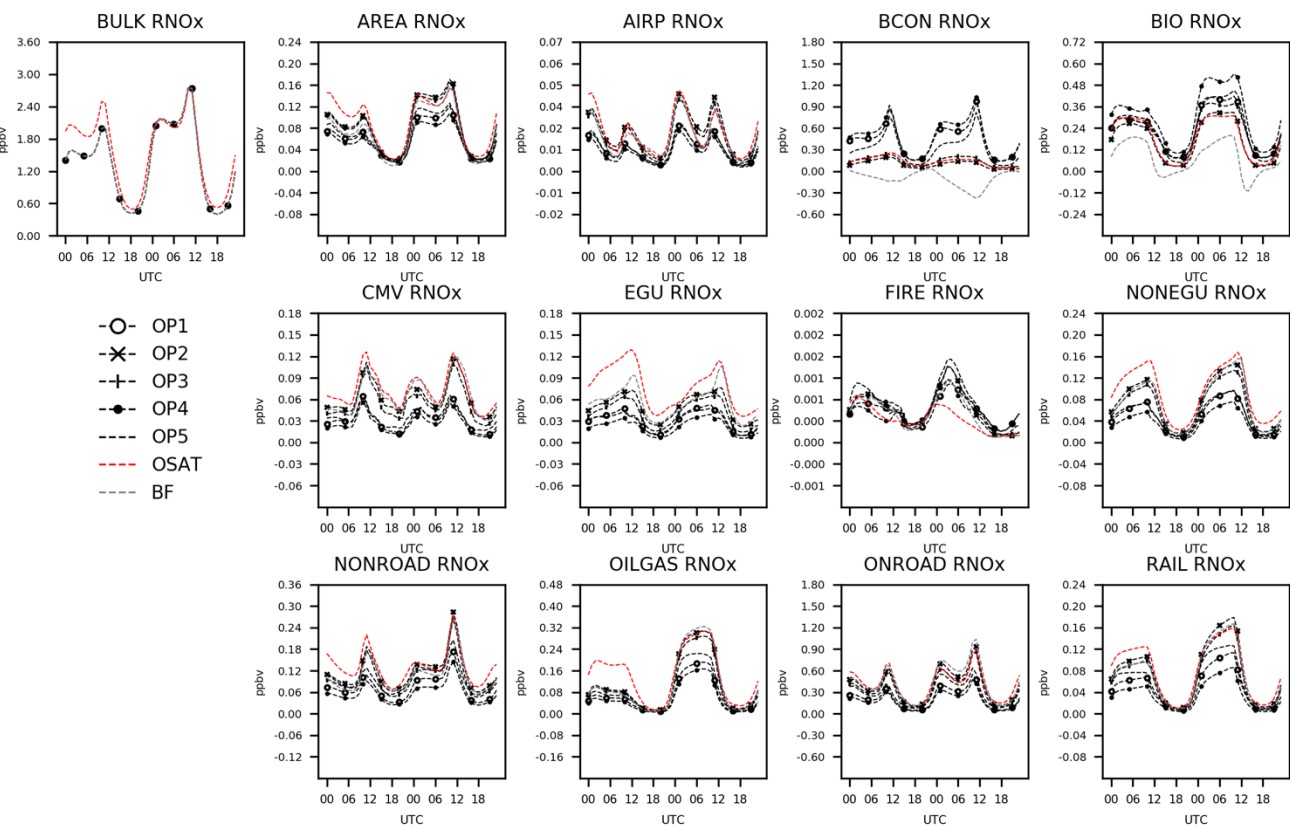


**Fig. 4 Total and attributed RNO$_x$ concentrations to various sectors as a function of hour of day and apportionment technique.**



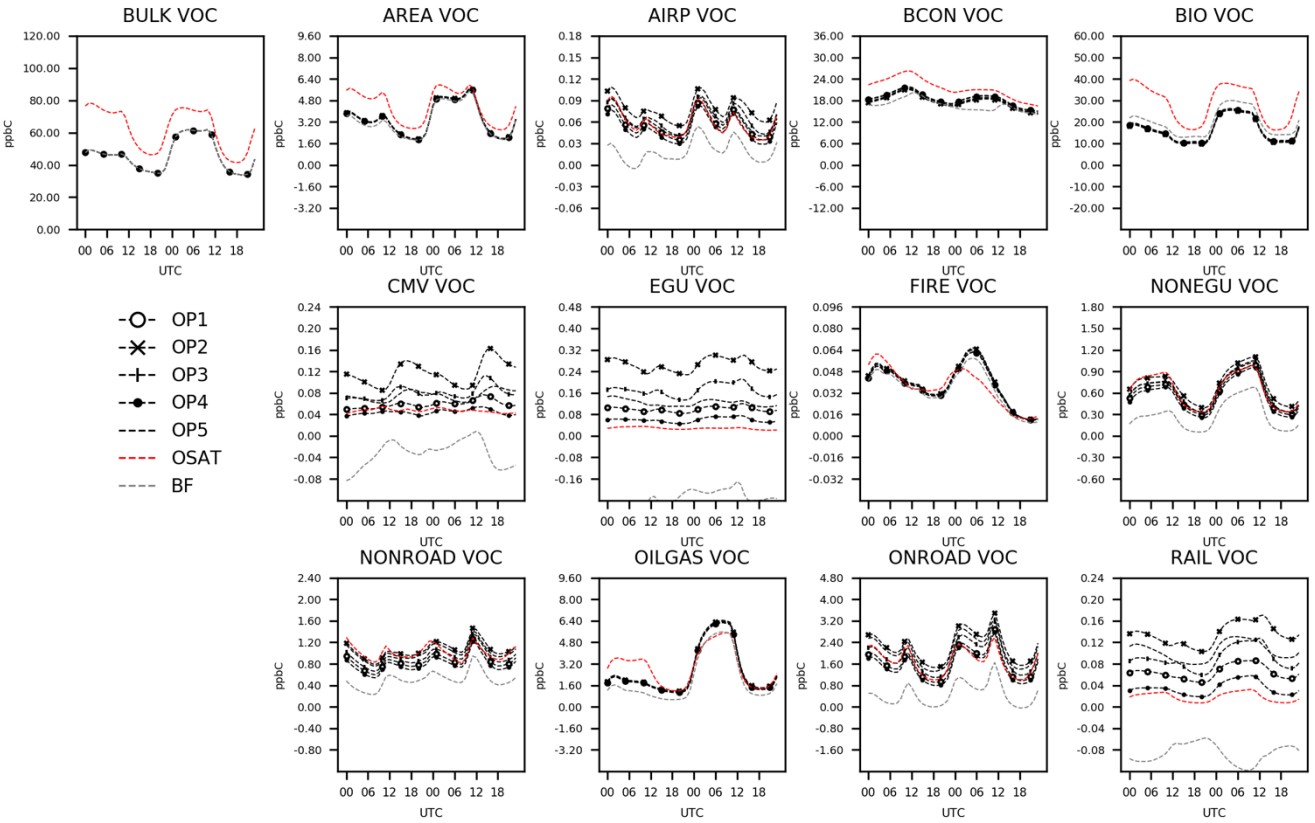

**Fig. 5 Total and attributed VOC concentrations to various sectors as a function of hour of day and apportionment technique.**


## 4.2.2 Spatial distribution of source apportionment simulations

Spatial patterns of total and sector contributions of MDA8 $O_3$ (Fig. 6), $RNO_x$ (Fig. 7) and VOC (Fig. 8) have been examined for the seven simulations. In Fig. 6, OSAT exhibits the same spatial distribution of MDA8 $O_3$ total concentrations as other CMAQ-based simulations (OP1, OP2, OP3, OP4,

OP5, and CMAQ-BF), with the exception of OSAT's relatively high marine and offshore total concentrations (5 ppbv), which could be explained by the difference in planetary boundary layer dynamics and gaseous chemical mechanism configuration between the two parent models. CMAQ CB6R3 uses a rough parameterization for full marine halogen chemistry to destroy $O_3$, depending only on land-use category and sunlight (Sarwar et al., 2015, 2019), whereas CAMx CB6R4 handles $O_3$

depletion in the marine boundary more efficiently by including the 16 most important reactions of inorganic iodine (I-16b, Emery et al., 2016). According to a sensitivity test conducted by Smith et al.





(2016), I-16b could reduce $O_3$ depletions by 2-5 ppbv in comparison to full halogen chemistry. Regarding sector concentrations, the spatial distributions of seven simulations are comparable. They can all capture geographic contribution hot spots from each sector, although their magnitudes vary. For most sources, OSAT paradoxically shows lower contributions over the ocean. OP2 stands out with fewer contributions from BIO than the other four ISAM options, and subsequently assigns larger concentrations to other sectors, particularly over east coastal regions, as shown in Fig. 3 and 6. Since OP2 assigns all products to sources with nitrogen reactants, the influence of reactants from biogenic sources is diminished, as intended.

Figure 7 depicts the associated outcomes of $RNO_x$. Except for BCON, the seven simulations produce geographically and quantitatively consistent findings. From the spatial distributions, we can conclude that local sources govern $RNO_x$ more than long-transported sources compared to $O_3$. Anthropogenic $RNO_x$ is either more concentrated in the urban areas (e.g., AREA, NONEGU, NONROAD), gasoline industry (OILGAS) and electric facilities (EGU) or along with transportation (e.g., AIRP, ONROAD, CMV and RAIL). Biogenic $RNO_x$ is more prevalent in rural locations with vegetation. It should be noted that OP1, OP4 and OP5 show more BCON $RNO_x$ across the entire domain because of the way to assign products in nitrogen related reactions (Section 2). OP1, OP4 and OP5 show local hotspots of $RNO_x$ attributed to BCON. Since there is no physical reason to suspect hotspots over urban areas, we conclude that these contributions represent $RNO_x$ attributed based on VOC or oxidants transported from the boundary. Figure 8 depicts the outcomes associated with VOC. Higher VOC concentrations from CAMx already shown in Fig. 6 are primarily from Virginia and North Carolina (OSAT bulk). As CMAQ and CAMx both use the same BEIS inventory data, the difference in total VOC concentrations may result from other differences between two models, like chemistry or deposition, accordingly, leading to higher biogenic sources in CAMx (BIO). For the rest of sectors, OSAT and ISAM options are fairly consistent except that the OP2 predicts more contributions from EGU, CMV and RAIL. CMAQ-BF predicts consistently lower source contributions for MDA8 $O_3$, $RNO_x$, and VOC, as shown in Section 4.2.1. This yet again illustrates that brute force represents an integrated sensitivity while the OSAT and ISAM represent attribution at a point in the nonlinear chemical systems. Monthly averaged spatial maps for MDA8 $O_3$, $RNO_x$, and VOC are also included in





425     Fig. S4(a-c) and show consistent results as two-day averaged maps. This demonstrates that our case study is appropriate, efficiently selecting representative days as well as minimizing the uncertainties from parent models (CMAQ and CMAQ). Additional figures of other grouped nitrogen species tracked in Table 4 (e.g., RGN, $HNO_3$ and $NO_y$) can also be found in SI.





430





**Fig. 6 Spatial comparisons of seven simulations for two-day averaged O$_3$ (08/09 and 08/10).**





435



**Fig. 7 Spatial comparisons of seven simulations for two-day averaged RNO$_x$ (08/09 and 08/10).**











**Fig. 8 Spatial comparisons of seven simulations for two-day averaged VOC (08/09 and 08/10).**

## 5 Model Simulation Time

The CPU time required to complete a source apportionment simulation in a 3D AQM is an important consideration for usability. For a 4 km x 4 km simulation domain encompassing the northeast U.S., the model run times for OSAT and ISAM are similar. Using 128 processors, base CMAQ (without ISAM) and CMAQ-ISAM simulations (11 source categories) are tested. Base CMAQ requires around



60 minutes per simulation day (24 hours), whereas CMAQ-ISAM requires approximately 120 minutes. If the number of processors is increased to 256, the simulation time for CMAQ-ISAM can be reduced to 30 minutes showing good scalability. It is worth noting that our CMAQ-ISAM simulations simultaneously track all additional species classes, such as sulfate, nitrate, ammonium, elemental

carbon, organic carbon, and chloride. It would shorten simulation times if related species were only tracked for $O_3$. Base CAMx (without OSAT) and CAMx OSAT are also tested with 128 processors, taking 37 and 67 minutes, respectively. CAMx also provides an optional tool for particles that can be simultaneously applied similarly to ISAM (PSAT, Yarwood et al., 2007). When additional pollutants are selected for tracking (e.g., sulfate, primary $PM_{2.5}$ species, etc.) total simulation time will increase for

both ISAM and OSAT/PSAT. CMAQ-BF speed is based on CMAQ base simulation (60 mins/day x (1 base + 11 sources + 1 boundary condition + 1 initial condition + 1 other) = 720 mins/day).

## 6 Discussions and Conclusions

Source attribution approaches are generally intended to determine culpability of precursor emissions sources to ambient pollutant concentrations. Source-based apportionment approaches such as

ISAM and OSAT provide similar types of information, specifically an estimate of which sources or groups of sectors (e.g., a sector) contributed to the air quality measured or estimated at a particular location. The assumptions in each technique have implications for interpretation in the context of air quality management.

Source attribution of secondarily formed pollutants cannot be explicitly measured, which makes

evaluation of source apportionment approaches challenging. Here, the ISAM approach was evaluated by 1) a comparison with a source apportionment approach implemented in a different photochemical modeling system and 2) a comparison with a simple source sensitivity (brute-force difference) approach in the same modeling system that is most comparable to source apportionment in more linear systems and less useful when formation and transport is nonlinear. Further, this section notes qualitative

consistency between the spatial nature of sector emission and the attribution of precursors and $O_3$ as another method to generate confidence in these approaches.





In this study, multiple apportionment approach comparisons show common features but still reveal wide variations in predicted sector contribution and species dependency. The attribution to sources emitting $NO_x$ and VOC is consistent with the spatial nature of these sources, which provides

confidence in the approach. However, nitrogen species (e.g., $NO_x$), for instance, are more sensitive to the choice of ISAM options than VOC. For example, although the attribution of $NO_x$ to EGUs matches the location of these sources (e.g., New York urban area) for all ISAM options, OP1, OP4 and OP5 predict more BCON $NO_x$. This is because the fast $NO_x$ cycling process assigns anthropogenically emitted nitrogen species to other sources, as the original emitted source identity is not retained through

these complex reactions. Further, sources entirely located offshore, such as commercial marine vessels, do not have culpability assigned to distant inland regions of the model domain. Most of the time, the amount of attribution to a certain sector depends on the number of emissions from that sector, how far away those emissions are, and whether the prevailing winds carried emissions from those places to the monitor or grid cell where air quality was predicted.

Among all ISAM options, the OP5 option, after making the assignment decision based on the ratio of $PH_2O_2$ to $PHNO_3$, is expected to predict generally similar spatial and temporal patterns to the OSAT source apportionment approach implemented in CAMx. However, it still shows disparity for some sectors (e.g., biogenic sectors for $O_3$). This result may be because of the OSAT formulation which differs from the ISAM options presented here. The OP5 option was also similar to brute-force

sensitivity estimates predicted in CMAQ with the exception of source groups that dominate regional emissions or $O_3$, such as biogenic VOC and $O_3$ introduced into the model through boundary inflow. In those situations, it is not reasonable to expect a source sensitivity approach to provide a useful comparison for source attribution given the highly nonlinear change in atmospheric chemistry.

By comparing the multiple approaches in the Northeast U.S., we found that both OSAT and

ISAM attribute the majority of $O_3$ and $NO_x$ contributions to boundary, mobile, and biogenic sources, whereas the top three VOC contributions are attributed to biogenic, boundary, and area sources. However, it is worthwhile to note that our results in this study are based on limited duration and specific regions, and they may not comprehensively reflect all situations. We continue to need further efforts

that combine field experiment studies and model evaluations for longer terms and multiple regions to

better understand source attribution given the highly nonlinear change in nature of $O_3$-$NO_x$ chemistry.

**Code availability**

The CMAQ model documentation and released version 5.4 of the source code, including updated ISAM code used in this study, are available at www.cmaq-model.org. The updates described here, as well as model post-processing scripts, are available upon request.

**Data availability**

The raw observation data used are available from the sources identified in Sect. 3, while the post-processed observation data are available upon request. The CMAQ model data utilized are available upon request as well. Please contact the corresponding author to request any data related to this work.

**Author contributions**

QS, SN, KB designed this study and experiments. QS led the development of this manuscript and was responsible for most of the model evaluation components in this study. SN, WH and BM developed the ISAM code. KB provided all the input data for the CMAQ simulations. QS carried out the CMAQ pre-processing, simulations, and post-processing, produced the figures, and prepared the initial paper draft. SN contributed directly to the writing of Sect. 2 of this paper. KB contributed directly to the writing of

Sect. 6 of this paper. WH, BM, CH and BH discussed all results throughout the ISAM development and contributed to the final writing of this paper.

**Competing interests**

The authors declare that they have no conflict of interests.



## Disclaimer

The views expressed in this article are those of the authors and do not necessarily represent the views or policies of the U.S. Environmental Protection Agency.

## Acknowledgments

This project was supported in part by an appointment to the Research Participation Program at the Office of Research and Development, US Environmental Protection Agency, administered by the Oak

Ridge Institute for Science and Education through an interagency agreement between the US Department of Energy and the EPA.

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
