# Peer review of "Comparison of Ozone Formation Attribution Techniques in the Northeast United States"

_Geoscientific Model Development, 2022_

## Author Comment (AC1)

**Source Attribution of Ozone and Precursors in the Northeast U.S. Using Multiple Photochemical Model Based Approaches (CMAQ v5.3.2 and beyond)**

Qian Shu[1], Sergey L. Napelenok[1], William T. Hutzell[1], Kirk R. Baker[1], Benjamin Murphy[1], Christian Hogrefe[1], Barron H. Henderson[1]

[1]U.S. Environmental Protection Agency, Research Triangle Park, NC, 27711, USA.

Correspondence to: Sergey L Napelenok (sergey.napelenok@epa.gov)

**Response to reviewer 1**

**https://gmd.copernicus.org/preprints/gmd-2022-273#RC1**

**General comments**

**The focus of this manuscript is a comparison of 5 versions of the Integrated Source Apportionment Method (ISAM) in the Community Multiscale Air Quality Model (CMAQ), the Ozone Source Apportionment Technology (OSAT) in the Comprehensive Air-quality Model with Extensions (CAMx), and the brute force method in CMAQ. This is a subject of interest to the audience of Geoscientific Model Development. To compare the source apportionment methods, the authors chose the model configurations to be as close as possible, with one exception, and picked days when the MDA8 O3 predictions of the models agreed well. There are numerous results in the manuscript and the supplement. The manuscript is reasonably well written, with some exceptions. A minor exception is that the reference list needs attention; some citations in the text are not in the list and vice versa.**

Thank you for the helpful comments. Each comment is addressed individually below and highlighted in blue. All these changes have been made in our revised version of manuscript and highlighted in yellow.

**I have two major issues with the manuscript. One is that Section 2 does not adequately describe the new/updated versions of ISAM in enough detail to be understood, nor does this section compare the ISAM versions to OSAT in detail so that the reader can understand the differences between all the methods. The authors should take 2 or 3 reactions of different types and explain, using equations, how the products are allocated to sources and how the allocations propagate to allocation of O3 formation if O3 is not a direct product of the reactions. Given that the authors have submitted the manuscript to Geoscientific Model Development, the readers should be informed of the details of the source allocation methods, to the point that someone could implement such methods in other models. That is a major value of this journal.**

We agree with you that we should clarify our ISAM updates. We have added more details and revised several places, as below. OSAT have been well-documented in the CAMx user guide (p. 173–p. 178, https://camx.com/Files/CAMxUsersGuide_v7.10.pdf) and previous work (Kwok et

al., 2015). We summarized OSAT and expanded the description of the ISAM updated in Section 2 as per your suggestion. The revised text for section 2 is as follows.

For ISAM (Lines 149-190):

"The existing scheme based on stoichiometrically proportional product attribution introduced in CMAQ version 5.3.2 has been retained as ISAM option 1 (ISAM-OP1). Four new options have been added so the user can configure their simulation based on the application's goal. Each option allows for greater retention of source identity based on subsets of species in the chemical mechanism. ISAM-OP2 apportions products according to the source identity of reactive nitrogen species, including NO, NO2, nitrate radical (NO3), nitrous acid (HONO), HNO3, dinitrogen pentoxide (N2O5), and aerosol nitrate (ANO3). For example, CB6R3 contains the following reaction between the methyl peroxy radical (MEO2) and NO:

MEO2 + NO = FORM + HO2 + NO2 (R2)

In the original ISAM-OP1 configuration, the products of R2, FORM, HO2, and NO2 inherit source identities proportional to the source identities of the reactants (MEO2 and NO). However, ISAM-OP2 apportions the product to be from the source identity of NO (presumed predominantly anthropogenic), because NO is a weighted nitrogen-containing species. When a reaction's reactants do not include any of the weighted species, products are apportioned to source identities using the same methodology used in OP1.

ISAM-OP3 expands OP2's list of weighted species to include VOC species identified as important to O3 production. In CB6R3, this includes aldehydes (ALD2 and ALDX), FORM, acetone (ACET), lumped ketones (KET), peroxy operators (XO2 and XO2H), ISO2, acetyl peroxy radicals (C2O3 and CXO3). Therefore, products of reactions containing these VOCs in addition to the nitrogen species of OP2 as reactants would inherit these species' source identities. For example, ALD2 reacts with the NO3 as follows in CB6R3.

ALD2 + NO3 = C2O3 + HNO3 (R3)

The reaction's products, C2O3 and HNO3, inherit identities equally divided between the sources of the reactants because ALD2 and NO3 are on the list of OP3 species. Reactions without any of these species in the reactants list, like OP2, have their products apportioned to source using OP1's methodology when the reactants are not among the weighted ones.

ISAM-OP4 lists only VOC species and daughter products instrumental in O3 chemistry as defined in OP3. In the R1 example, the products are apportioned to the source identity of ISO2, because the other reactant, NO, is not on the list of weight species. Similarly, the products of R3 are attributed to the source identity of ALD2. As in options 2 and 3, reactions (such as R2) without any listed species are attributed as in OP1's method.

Finally, ISAM-OP5 was added to account for the instantaneously calculated O3 formation regime or limiting case. The regime is determined using the ratio of PH2O2/PHNO3. The transition point between regimes has a default value equal to 0.35 (Sillman, 1995). For the NOx-limited regime (PH2O2/PHNO3>0.35), source identity is passed from the nitrogen species of OP2, while for the VOC-limited regime (PH2O2/PHNO3≤0.35), source identity is passed from the organics of OP4. These CMAQ-ISAM options, including the regime threshold value (or transition point), are accessible at runtime through the standard model run script.

Table 1. Expanded CMAQ-ISAM options.

| CMAQ ISAM option | Reaction product source identity assignment | Representative CB6R3* Species |
|---|---|---|
| ISAM-OP1 | Proportional to stoichiometry of all reactants. | All tracked model species |
| ISAM-OP2 | Proportional to stoichiometry of nitrogen containing reactants, otherwise same as ISAM-OP1. | NO, NO2, NO3, HONO, HNO3, N2O5, ANO3 |
| ISAM-OP3 | Proportional to stoichiometry of key O3 chemistry reactants (reactive VOCs, radicals, nitrogen species), otherwise same as ISAM-OP1. | NO, NO2, NO3, HONO, HNO3, N2O5, ANO3, ALD2, ALDX, FORM, ACET, KET, XO2, XO2H, ISO2, C2O3, CXO3 |
| ISAM-OP4 | Proportional to stoichiometry of VOC and radical containing reactants, otherwise same as ISAM-OP1. | ALD2, ALDX, FORM, ACET, KET, XO2, XO2H, ISO2, C2O3, CXO3 |
| ISAM-OP5 | According to the ratio of PH2O2 to PHNO3 if O3 chemistry reactants present, otherwise same as ISAM-OP1. | NOx-limited: NO, NO2, NO3, HONO, HNO3, N2O5, ANO3 VOC-limited: ALD2, ALDX, FORM, ACET, KET, XO2, XO2H, ISO2, C2O3, CXO3 |

*Species are based on CB6R3 and may vary based on different chemical mechanisms implemented in CMAQ. Details can be found in SA_DEFN.F in the CMAQ source code.

For OSAT (Lines 191-211):

"The source apportionment approach implemented in CAMx is briefly recapped here. Detailed updates of all OSAT versions can be found in CAMx official user guide (https://camx.com/Files/CAMxUsersGuide_v7.10.pdf). All available versions of OSAT (including OSAT3) in CAMx separately solve for production and destruction of O3 with production being attributed to either NOx or VOC emissions, depending on which is estimated to be limiting O3 production. When the ratio of PH2O2/PHNO3 exceeds 0.35, the produced O3 is attributed to NOx emissions, and VOC emissions below that threshold. The CAMx source apportionment implementation includes an option (OSAT-APCA) that allows for a redirection of attribution to anthropogenic emissions in situations where the limiting precursor is biogenic. In CAMx-OSAT, O3 attributed to NOx and VOCs is tracked as separate tracer groups. O3 tracers are first adjusted to account for O3 destruction processes and subsequently for net O3 production, which is defined as the difference between O3 production and O3 destruction based on a subset of photochemical reactions that result in O3 destruction. In situations where the net O3 production is negative (destruction reactions dominate), all the O3 tracers are proportionally decreased. When net O3 production is positive, production is assigned proportionally to the sources of those emissions (NOx and VOC precursor tracers) at the time and place where O3 was made. OSAT includes a group of tracers that track odd-oxygen that is consumed when O3 reacts with NO to form NO2 that can quickly photolyze and reform O3 through a reaction with oxygen. In this situation, the O3 removed from the O3 tracers due to the NO + O3 reaction is moved to the odd-oxygen tracers (which have separate NOx and VOC tracer groups). When NO2 is photolyzed

*and O3 formed a proportional amount of O3 is taken from the odd-oxygen tracers and moved to the O3 tracers."*

**My other major issue is that the authors used a different chemical mechanism for the ISAM and OSAT simulations. Because source apportionments depend on the chemistry used, this is a significant limitation of the work and reduces its value to readers. As the authors note, differences in the source apportionments could be due to differences between ISAM and OSAT or due to differences in the chemical mechanisms in the two models or both. Some of the differences in the source apportionments are puzzling, suggesting that the difference in the chemistry could be important. Because the chemistry is different between CMAQ and CAMx, the conclusions of the manuscript are rather vague, e.g., lines 19-23. Consequently, the authors provide little guidance on which ISAM methods should be used and for what purpose.**

Evaluating the accuracy of source apportionment model results is challenging because the source contribution of secondary pollutants such as ozone cannot be assessed independently based on observations. In this case, we use CAMx-OSAT and brute-force methods as alternative references. The primary objective of this paper is to document recent ISAM updates and demonstrate their impacts on source apportionment results for O3 and its precursors for added ISAM options. OSAT and ISAM are two different source apportionment methods, embedded in the two different parent models, CMAQ and CAMx. We are not making a strictly consistent comparison because that is impossible, considering there are many differences in model formulations and data requirements. However, we have tried to make most configuration options as similar as possible. Chemical mechanism is one of the things that we can't resolve perfectly, as it is not feasible to use the same version of chemical mechanism between CMAQ v5.3.2 and CAMx v7.10. The most updated carbon bond mechanism in CMAQ v5.3.2 is CB6R3 (https://github.com/USEPA/CMAQ/blob/5.3.2/CCTM/src/MECHS/README.md) while CAMx v7.10 has CB6r2h and CB6r4/r5 as Table 5-1 in the CAMx user guide(https://camx.com/Files/CAMxUsersGuide_v7.10.pdf). CMAQ has an alternative chemical mechanism called "CB6R3m" that adds detailed halogen chemistry and DMS. Sarwar et al. (2015, 2019) demonstrated that updating CB6R3m is more beneficial in the hemispheric CMAQ model, where the influence on intercontinental transport over oceans is larger than over land. Model sensitivity runs were also completed with CB6R3 (without detailed halogen and DMS chemistry) and CB6R3m (with detailed halogen and DMS chemistry) over the Northern Hemisphere for three months in 2015 (October–December) by Sarwar. It reduces ozone by 3–14 ppb (Figure 1) over much of the ocean. It reduces ozone over land by much smaller margins than over sea water (https://github.com/USEPA/CMAQ/blob/5.3.2/DOCS/Release_Notes/detailed_halogen_and_DMS_chemistry.md).

For this study, as our focus was more on the regional domain over the Northeast U.S., CB6R3 was chosen for CMAQ-ISAM. It is noteworthy that the major updates for CB6R4 from CB6R3 are to (1) replace full marine halogen chemistry with a condensed iodine mechanism called "I-16," which could reduce ozone depletion over marine areas, and (2) add dimethyl sulfide (DMS) chemistry. Emery et al. (2016) demonstrated that the difference in ozone decrements between full halogen chemistry and I-16 is small and can be neglected over land. In this case, CB6R4 was

chosen rather than CB6R2h and CB6R5. With these two chemical mechanism configurations, our study shows similar results to Sarwar et al. (2015, 2019) and Emery et al. (2016) when CMAQ predicted total MDA8 O3 compared to that of CAMx (Figure 6 in the paper). We have discussed these in Lines 390–396. Although we cannot eliminate the influence of different chemical mechanisms, just like other potential uncertainties, we tried to diminish the inevitable difference in this study. It is still valuable to show these intercomparisons between ISAM and OSAT at some levels. Future studies could be done when two models implement an identical chemical mechanism.

We have also added some lines to clarify it. Lines 232-236 "*It is noteworthy that the major updates for CB6R4 from CB6R3 are to (1) replace full marine halogen chemistry with a condensed iodine mechanism called "I-16," which could reduce $O_3$ depletion over marine areas, and (2) add dimethyl sulfide (DMS) chemistry. Emery et al. (2016) demonstrated that the difference in $O_3$ decrements between full halogen chemistry and I-16 is small and can be neglected over land.*"

[Figure]

Figure 1: Impact of halogen chemistry on ozone (three-month average).

**Table 5-1. Gas-phase chemical mechanisms currently implemented in CAMx v7.1.**

| Mechanism ID | Description |
|---|---|
| CB05 | Carbon Bond 2005 (Yarwood et al., 2005b). 156 reactions among 51 species (38 state gases, 13 radicals). |
| CB6r2h | Carbon Bond v6, "Revision 2" (Yarwood et al., 2010; Yarwood et al., 2012a; Hildebrandt Ruiz and Yarwood, 2013), with updates to include reactions involving oceanic halogen compounds (Yarwood et al., 2014). 304 reactions among 115 species (88 state gases, 27 radicals). |
| CB6r4 | Carbon Bond v6, "Revision 4" adds temperature- and pressure-dependent $NO_2$-organic nitrate branching from CB6r3 (Emery et al., 2015), a condensed set of reactions involving ocean-borne inorganic iodine (Emery et al., 2016a), and DMS oxidation reactions (Emery et al., 2019). 233 reactions among 87 species (62 state gases, 25 radicals). |
| CB6r5 | Carbon Bond v6, "Revision 5" incorporates recent updates to chemical reaction data from IUPAC (Atkinson et al., 2004; IUPAC, 2019) and NASA (Burkholder et al., 2015) for inorganic and simple organic species that play a role in ozone formation. Same number of reactions and species as CB6r4. |
| SAPRC07TC | The Statewide Air Pollution Research Center 2007 mechanism that includes updates to support toxics and numerical expressions of rate constants to support the current chemistry mechanism compiler (SAPRC07TC; Carter, 2010; Hutzell et al., 2012). 565 reactions among 117 species (72 state gases, 45 radicals). |
| MECH10 | A user-defined simple chemistry mechanism can be developed for any gas and/or particulate species, which is defined by a "Mechanism 10" parameters file and solved within a user-supplied subroutine called "chem10.f." |

**Specific comments**

**Line 59. $NO_x$ (as $NO_2$) also removes OH to $HNO_3$, and this is usually a greater impact on the $O_3$ formation than the titration of $O_3$. Titration produces $NO_2$, which can quickly photolyze and produce $O_3$ again, but OH loss slows $O_3$ formation for an extended time period.**

We agree with your comment. We did not express it very clear. We have rewritten it in Lines 59-62 "*For example, removing $NO_x$ may lead to an increase of $O_3$ concentrations in the vicinity of large NO emissions (e.g., power plants), as the result of net conversion of $O_3$ to $NO_2$ (Gillani et al., 1996) or at night-time when $NO_x$ titration cannot be balanced by the photolysis of $NO_2$.*"

**Line 95. Are the updates just changes to flexibility in application or do they also include more substantial changes that affect how $O_3$ is apportioned to sources?**

The original ISAM implementation is retained as OP1. Additional apportionment options are added as OP2-5. These new options do increase flexibility for the user, who can now make a choice to as to which is most appropriate for their application. The new options can also substantially change apportionment results as is documented in our study. These differences stem from model algorithms attributing reaction product based on chemical compositions of

reactants and are detailed (hopefully more clearly) in Section 2. Therefore, the updates both increase flexibility and include substantial changes to ozone apportionment.

**Lines 150-152.  Unclear.  How is ISAM-OP3 different from ISAM-OP1?**

OP3 distributes product towards sources with reactants that are typically associated with ozone chemistry. While these the majority of species in a given chemical mechanism do, to some extent, participate in ozone chemistry, it is not all species (which is what is included in OP1). We have rewritten the ISAM option descriptions in section 2 to address this one and previous concerns from the reviewers in more detail.

**Lines 154-158.  Is ISAM-OP5 the same as OSAT3?  If not, what are the differences?**

They are not the same. In general, ISAM (all options) tracks individual species based on selected chemical mechanisms. The source apportionment of these tracked species is based on integrated reaction rates and product yields. ISAM-OP5 is designed to better understand ozone attribution but can also be used for other species. For O3 attribution, it tracks either related nitrate species or VOC species according to the ratio of PH2O2 to PHNO3 if ozone chemistry reactants are present in related reactions. When ISAM-OP5 is used to assign O3 sources, there are a total of 55 tracers for each source tag for a single domain across the entire chemical reactions in CB6R3. However, according to Table 1, only a subset of species, including either reactive nitrogen species (NO, NO2, NO3, HONO, N2O5, and ANO3) or VOCs that are important to O3 production (ALD2, ALDX, FORM, ACET, KET, XO2, XO2H, ISO2, C2O3, and CXO3), will be used to assign sources when they are present in reaction reactants.

CAMx-OSAT is designed for ozone source apportionment. As we have recapped in Lines 199–206, "*In CAMx-OSAT, O3 attributed to NOx and VOCs is tracked as separate tracer groups. O3 tracers are first adjusted to account for O3 destruction processes and subsequently for net O3 production, which is defined as the difference between O3 production and O3 destruction based on a subset of photochemical reactions that result in O3 destruction. In situations where the net O3 production is negative (destruction reactions dominate), all the O3 tracers are proportionally decreased. When net O3 production is positive, production is assigned proportionally to the sources of those emissions (NOx and VOC precursor tracers) at the time and place where O3 was made.*" Unlike ISAM, OSAT3 is based on a subset of photochemical reactions, and the chemical conversion pathways between CB6 and OSAT3 are summarized in the figure below from the CAMx user guide (p. 176). OSAT3 tracks 10 families of tracers for ozone source apportionment based on its formulation (details on the tracers in OSAT3 can be found in my response to your following question).

[Figure]

Figure 7-5. Correspondence between NOy species in CB6 and tracer families in OSAT3 with conversions between species/tracers shown by arrows.

**Lines 155-158. This seems in conflict with Table S3. For OP5, Table S3 indicates that the PH2O2/PHNO3 ratio affects whether or not O3 is allocated to VOCs but that O3 is allocated to NOx species whether the ratio is above or below the 0.35 threshold.**

We have rewritten the ISAM option descriptions. We also removed Table S3, which could confuse readers, and replaced it with more explicit tracked species information for each ISAM option in Table 1.

**Line 160, Table 1, ISAM-OP1. Is the source attribution based on the reaction products or the reactants? Lines 115-125 suggest it is the reactants. Also, for ISAM-OP5, the Table mentions ISAM-OP3 but lines 155-156 mention ISAM-OP2 and ISAM OP4. Confusing**

**and unclear.**

All five options are based on reactants. We have removed Table S3 and rewritten ISAM descriptions. In Lines 180-184 "*Finally, ISAM-OP5 was added to account for the instantaneously calculated $O_3$ formation regime or limiting case. The regime is determined using the ratio of $PH_2O_2/PHNO_3$. The transition point between regimes has a default value equal to 0.35 (Sillman, 1995). For the $NO_x$-limited regime ($PH_2O_2/PHNO_3>0.35$), source identity is passed from the nitrogen species of OP2, while for the VOC-limited regime ($PH_2O_2/PHNO_3\leq0.35$), source identity is passed from the organics of OP4.*", we explained ISAM-OP5 should switch between OP2 and OP4 based on the ratio of PH2O2/PHNO3.

**What is the total number of tracers used in the different versions of ISAM and in OSAT3?**

ISAM uses individual tracers for each tracked species, and the number of tracers depends on how many emissions sectors, regions, or species the user is tracking. OSAT3 uses family tracers instead of individual tracers. Detailed OSAT3 formulation is well documented in the CAMx user guide (p. 173–p. 178, https://camx.com/Files/CAMxUsersGuide_v7.10.pdf).

We use the CB6R3_AE6_AQ7 chemical mechanism as an example in CMAQ. When the O3 tag class is used in ISAM for a single source sector tag (e.g., EGU) in a single domain, a total of 55 tracers, including all related O3, nitrate, and VOC species, will be tracked for all five ISAM

options. (HO2, O3, O1D, O, O3P, ANO3(I,J), HNO3, NO, NO2, NO3, HONO, N2O5, PAN, NTR1, NTR2, INTR, PNA, PANX, CLNO2, CLNO3, XO2, XO2H, XO2N, ROR, MEO2, ISO2, C2O3, CXO3, CO, ALD2, ALDX, ETH, ETHA, ETOH, FORM, IOLE, ISOP, MEOH, OLE, ECH4, PAR, TERP, TOL, XYLMN, NAPH, ETHY, PRPA, ACET, KET, GLY, BENZENE, APIN, GLYD, MEPX). However, according to Table 1, the way of assigning sources to reactants is different for each ISAM option.

Based on p. 176–177 in the CAMx user guide, OSAT3 uses the following 10 tracers for each source group and region:

$V_i$ — VOC emissions
$NIT_i$ — Nitric oxide (NO) and nitrous acid (HONO) emissions

$RGN_i$ — Nitrogen dioxide ($NO_2$), nitrate radical ($NO_3$) and dinitrogen pentoxide ($N_2O_5$)
$TPN_i$ — Peroxyl acetyl nitrate (PAN), analogues of PAN and peroxy nitric acid (PNA)
$NTR_i$ — Organic nitrates ($RNO_3$)
$HN3_i$ — Gaseous nitric acid ($HNO_3$)
$O3N_i$ — Ozone formed under NOx-limited conditions from $N_i$
$O3V_i$ — Ozone formed under VOC-limited conditions from $V_i$
$OON_i$ — Odd-oxygen in $NO_2$ formed from $O3N_i$
$OOV_i$ — Odd-oxygen in $NO_2$ formed from $O3V_i$

**Are any of the ISAM versions close to or the same as OSAT3? What are the differences between ISAM-OP5 and OSAT3?**

All ISAM versions are not the same as OSAT3. However, ISAM-OP5 is designed for ozone source apportionment by accounting for the instantaneously calculated $O_3$ formation regime or, in the limiting case, using the ratio of PH2O2/PHNO3, similar to OSAT3. However, they still have different approaches. The difference between ISAM-OP5 and OSAT3 has been discussed in our previous response to your question in Lines 154–158.

**Lines 178-185. The authors did a good job of making most configuration options as similar as possible between the models and picking days when the MDA8 $O_3$ predictions of the models agreed well. However, CMAQ used CB6r3 and CAMx used CB6r4. The chemistry is a key driver of the source apportionments, and thus the apportionments will depend on the chemical mechanism. The CMAQ and CAMx simulations should have been done using the same chemical mechanism to eliminate the differences in chemistry as a possible explanation for the differences in the source apportionments. To make this paper useful to the modeling community, one set of simulations should be redone with the chemical mechanism used for the other set of simulations.**

Thanks for your suggestions. While it would absolutely be valuable to perform this study with and exactly consistent chemical mechanism, it was not possible for our case as described above (Our response to your second major concern).

**Lines 194-215. The description of OSAT3 should be in Section 2, after the description of ISAM. Pick the same 2 or 3 reactions used to give the details of ISAM and give the corresponding details of OSAT3.**

We have moved it to Section 2 as your suggestions (Lines 190-211).

**Lines 225-235. It is unclear what was done with the OTHR category. Line 226 states that there was a tracer for OTHR but lines 226-227 imply that OTHR was not tagged. Also, why cannot a BF simulation be done removing just the OTHR emissions? Lastly, the OTHR emissions should be included in Table 3 to show how large they are compared to everything else.**

We have added one-line sentences for clarification as Lines 259-260 "==In this study, all emissions sectors were tracked as previously mentioned above for OSAT and ISAM==." The user can only tag a subset of emission sectors, and all left-over emissions sectors are treated as OTHR since they are not specifically tagged. In our study, we have tagged all emission sectors; thereby, emissions going to OTHR should be very small or close to zero theoretically. For CMAQ-BF, acquiring OTHR contributions means that we need to remove all emissions for the simulation, which is not an appropriate way because it could shift the chemical system very drastically, especially for secondary polluted species like $O_3$. Table 3 shows the emissions we input into the CMAQ simulation for this study.

**Line 240-241. "… ISAM tracks all individual oxidized nitrogen and VOC species, …". But the footnotes to Table 4 state that ISAM does not track INO3, OPAN and CRON. This seems to be a contradiction. Please revise or provide an explanation why this isn't a contradiction.**

My apologies for the inappropriate expression. We have rewritten this sentence in Line 273 to "==ISAM tracks individual oxidized nitrogen and VOC species based on selected chemical mechanisms in CMAQ==."

**Lines 243-244. "…the two models have distinct species representation." CB6r3 and CB6r4 have different species? ISAM and OSAT use different species? Please clarify and give examples.**

My apologies again for the inappropriate expression. We have rewritten this sentence in Line 275 "==some differences still exist since species representations between the two models are not completely the same==." And we think our response to your second major concern could address this comment. Also, Table 4 presents the difference in species representation between OSAT and ISAM.

**Line 259. Correlation of $O_3$ concentrations? Correlation of all species concentrations?**

We have rewritten in Line 292 "*We initially set the correlation relationship ($R^2$) criteria of maximum daily 8-hour averaged (MDA8) $O_3$*"

**Line 278. "…inconsistent predicted concentrations." Please explain further.**

We have rewritten in Line 311 "*their predictions do not agree well with each other, with a difference of MBs up to 8 ppbv.*"

**Lines 295-296. The MB and NMB differences diminish for MDA8 $O_3$ but increase for hourly $O_3$.**

Thanks for your suggestion. We have rewritten in Line 328 "*The differences of MB, NMB and $R^2$ between the two models also diminish for MDA8 $O_3$ but increase for hourly $O_3$ from the monthly episode to the two-day episode.*"

**Line 350. "typically negative." Typically smaller?**

We have removed "typically negative" in Line 383 to avoid confusions.

**Lines 355-356. "Except for …all sectors." This sentence is redundant with the following sentence.**

We have removed the redundant sentence.

**Lines 365-376. The results in Figure 5 raise the question of why there is such a large difference between ISAM and OSAT for the BULK VOC results (which are reflected in differences in the BIO VOC, BCON VOC and AREA VOC results), when there is much better agreement in Figures 3 and 4 for BULK O3 and BULK RNOx. The difference for BULK VOC needs further investigation and explanation because it suggests some important difference(s) in the formulation or implementation of ISAM and OSAT. An alternative explanation is that the difference is due to differences in the chemical mechanisms, which should be remedied by using the same mechanism in both models. Because the BULK O3 and BULK RNOx agree reasonably well, other explanations seem unlikely.**

First, we disagree that this is either a source apportionment formulation or a mechanism problem. Given that the difference is present in the BULK concentration, we disagree that this is likely a difference in the formulation of ISAM and OSAT. Given that the domain as a whole (not NYC) is likely NOx-limited, we disagree that the similarity in BULK O3 and BULK RNOx precludes another explanation. Both of these models have different internal treatments for advection and diffusion, which can impact surface-level concentrations and indirectly impact chemical reactions. As previously discussed, the primary distinction between CB6R4 and CB6R3 is halogen chemistry and the preference for DMS over marine. This difference should not be related to the gas-phase chemical mechanism. Another possible reason could be the difference in representation of these VOC species between two models. As OSAT only has pre-lumped VOC species, we make similar calculations for individual VOC species in CMAQ to match OSAT;

however, individual VOC species comparisons are not available for this study. It is hard to identify the uncertainty that causes this difference at the current stage. We decided to flag this result in Lines 403-408 "*Given that the difference is present in the total concentration, this is unlikely caused by different source apportionment formulation between CMAQ and CAMx. As CAMx only gives pre-lumped VOC, it is challenging to compare individual VOC species between CMAQ and CAMx to explain this difference at current stage. Another possible reasons to cause it could be that models have different internal treatments for advection and diffusion, which can impact surface-level concentrations and indirectly impact chemical reactions.*" and to highlight it in our conclusion for future investigation as it is outside the scope of this paper in Line 542-550 "*However, comparisons of OSAT and ISAM have some limits, especially when they are under the two different parent models, CAMx and CMAQ. Although we have put efforts into diminishing the differences between the two models by making most configuration options as similar as possible, some inevitable uncertainties cannot be eliminated at the current stage of this study (e.g., an imperfect match of chemical mechanisms, different internal treatments for advection, diffusion, and deposition processes). Further, it is also worthwhile to note that our results in this study are based on limited duration and specific regions, and they may not comprehensively reflect all situations. Given that the source attribution of secondary pollutants cannot be explicitly measured, these inter-comparisons between ISAM and OSAT are still useful for reference.*"

**Line 391.  What is this 5 ppbv?  The total (BULK) offshore O3 concentration is clearly above 5 ppbv.**

We have corrected it in Line 428 "*> 5 ppbv*"

**Line 392. "and gaseous chemical mechanism configuration between the two parent models".  The same mechanism should be used in both models to avoid this ambiguity.**

We have revised this confusing sentence to Line 429 "*different marine chemistry configuration*".

**Line 400.  "For most sources, OSAT paradoxically shows lower contributions over the ocean."  However, CAMx BULK O3 is larger than CMAQ BULK O3 over the ocean.  Assuming that OSAT BULK O3 is the sum of the contributions from the individual source categories, there must be enough increased marine O3 from some sources (e.g., BCON and EGU) that there is no inconsistency/paradox in the OSAT results.  The increased O3 is from a few sources, not distributed across all sources, which may be a consequence of the OSAT procedure for allocating O3.**

Thanks for clarifying. We have removed this sentence.

**Lines 411-415.  The BCON results using OP1, OP4 and OP5 are strange.  The authors' conclusion that the OP1, OP4, and OP5 results are due to VOC or oxidants transported from the boundary is not at all obvious.  This is especially true because the OSAT and BF results show little impact of BCON, and the only impact is very near the west boundary.  Again, without an understanding of how the products are allocated in ISAM**

**and the impact of the chemistry differences between CB6r3 and CB6r4, it is not possible to understand these BCON results. In addition, the OP1, OP4 and OP5 results for BCON raise the question of whether these versions of ISAM are useful.**

We have revised Section 2, and we believe it better represents the difference among the five ISAM options. Like what we have explained in our paper Lines 118-139 "*For example, the isoprene peroxy radical (ISO2) reacts with nitric oxide (NO) to produce several different stable and radical species as represented in the CB6R3 chemical mechanism by the following reaction R1.*

$$ISO2 + NO = 0.1*INTR + 0.9*NO2 + 0.673*FORM + 0.9*ISPD + 0.818*HO2 + 0.082*XO2H + 0.082*RO2 \text{ (R1)}$$

*In addition to nitrogen dioxide (NO2), the products include isoprene nitrate (INTR), formaldehyde (FORM), hydroperoxy radicals (HO2), alkoxy radicals (XO2H), peroxy radicals (RO2), and other isoprene reaction products (ISPD). ISO2 is a product of the oxidation of isoprene, which originates from overwhelmingly biogenic sources. NO is typically emitted from anthropogenic combustion processes, with a much smaller natural component originating from lightning strikes and microbial soil processes on the global scale (Jacquemin et al., 1990; Yienger et al., 1995). Thus, the reactants are approximately half from biogenic and half from anthropogenic sources, so the reaction's products have the same attribution distribution. However, source attribution approaches, both receptor-based (such as PMF) and source-based (such as ISAM), are often used to understand how originally emitted NOx and VOC from particular sources ultimately contribute to model-predicted O3 production. The loss of source identity through processes such as the NOx cycle and the role of organic peroxy radicals from sources not controlling O3 production make it difficult to determine the culpability of emission sources. In the preceding example, the NO2 produced by R1 is assigned a source that is approximately 50% biogenic and 50% anthropogenic. These source assignments propagate quickly when catalytic processes cause NO2 to cycle back to NO through photooxidation and radical oxidation Because NOx cycling is fast in regional air pollution models, anthropogenically emitted nitrogen species can be assigned to biogenic (or other nearby) sources downwind, so the original source identity was not retained.*" We used R1 as an example to explain why this could happen. Based on the design of ISAM options, approximate half of the NO2 could come from ISO2, and the other half could come from NO in OP1. ISO2 contributes 100% of the NO2 in OP4. NO2 sources will switch between OP2 and OP4 based on the ratio of PH2O2 to PHNO3. In Figure 7, BCON RNOx in OP1, OP4, and OP5 match the spatial distribution of total RNOx where hot spots are captured over high RNOx concentrations, indicating OP5 could switch more to OP4 over these locations. We expected OP2 and OP3 to produce comparable results to OSAT and BF because these two options were forced to assign more sources with tracked nitrate species in Table 1 as designed. It demonstrated that OP2 is more suitable for RNOx attributions, broadly, and also for those species that can quickly circulate. However, OP1, OP4, and OP5 can still be useful for other species, like the VOCs that we have presented in this paper or other primary pollutants. This is one of the reasons we added the ISAM flexibility for the user to select.

**Line 416. "Higher VOC concentrations from CAMx already shown in Figure 6 …". Figure 6 shows O3, not VOC. Should the reference be to Figure 5?**

We have corrected it in Line 452 "*Fig. 5*".

**Lines 417-419. "…may result from other differences between two models, like chemistry or deposition, …". Again, using different chemistry in the two models significantly limits the conclusions that can be obtained with these results, making the paper less valuable to readers and to regulatory officials. The differences between OSAT and ISAM for Bulk VOC and BIO VOC need better explanation.**

As we have discussed before, we are not making a strictly consistent comparison because that is impossible considering there are many differences in model formulations and data requirements. In this study, it is also not possible to make chemical mechanisms that are completely consistent between ISAM and OSAT. Both of these models have different internal treatments for advection and diffusion, which can impact surface-level concentrations and indirectly impact chemical reactions. The primary distinction between CB6R4 and CB6R3 is halogen chemistry and the preference for DMS over marine. This difference should not be related to the gas-phase chemical mechanism. Another possible reason could be the difference in representation of these VOC species between two models. As OSAT only has pre-lumped VOC species, we make similar calculations for individual VOC species in CMAQ to match OSAT; however, individual VOC species comparisons are not available for this study. It is hard to identify the uncertainty that causes this difference at the current stage. We have mentioned this imperfection in our conclusion in Lines 542-550 and are going to look for further investigations when two models implement an identical chemical mechanism. "*Comparisons of OSAT and ISAM have some limits, especially when they are under the two different parent models, CAMx and CMAQ. Although we have put efforts into diminishing the differences between the two models by making most configuration options as similar as possible, some inevitable uncertainties cannot be eliminated at the current stage of this study (e.g., an imperfect match of chemical mechanisms, different internal treatments for advection, diffusion, and deposition processes). Further, it is also worthwhile to note that our results in this study are based on limited duration and specific regions, and they may not comprehensively reflect all situations. Given that the source attribution of secondary pollutants cannot be explicitly measured, these inter-comparisons between ISAM and OSAT are still useful for reference.*"

**Lines 419-421. It is surprising that the VOC contribution depends very little on the ISAM version for most source categories, but OP2 gives a significantly greater VOC contribution for CMV, EGU, and RAIL than do the other methods. CMV, EGU, and RAIL are sources with small VOC emissions (Table 3). The results suggest that OP2 is not valuable for source apportionment for sources with small VOC emissions, certainly not to apportion VOC emissions to them.**

We think these significantly greater VOC contributions from CMV, EGU, and RAIL for OP2 are amplified by the small scale of concentrations. They are actually very small sources of VOC (under 1 ppbv) compared to other sectors. OP2 contributes similar domain-wide averaged CMV, EGU, and RAIL contributions as other options. All details have been included in Fig. S6(c) and Table S2(a-b) in the supplement.

**Lines 455-456. If the CMAQ-BF time is equal to the quantity in parentheses, 60 mins/day X 15, the total should be 900 mins/day.**

We have corrected it in Line 492.

**Lines 476-480. Why does OSAT, which the authors expect to be most similar to OP5 (lines 485-487), give such a smaller contribution of BCON to RNOx than OP5 (and OP1 and OP4) in Figure 7? Does OSAT retain the emitted source identity through fast NOx cycling? The fact that OP1, OP4, and OP5 assign so much RNOx to BCON compared to the BF RNOx results suggest that these ISAM versions are not very accurate for RNOx.**

First of all, each ISAM option is different based on a different source assignment, as described in Table 1. We were expecting OP5 to be similar to OSAT for O3, but not for RNOx. We have clarified it in Lines 523-524 "*Among all ISAM options, the OP5 option, after making the assignment decision based on the ratio of $PH_2O_2$ to $PHNO_3$, is expected to predict generally similar spatial and temporal patterns for $O_3$ to the OSAT source apportionment approach implemented in CAMx.*" We also added one sentence in Lines 521–522. "*The designed five ISAM options maximize its flexibility, particularly for modeling source apportionment of $O_3$ and its precursors, but the choice of option depends on target species*" to emphasize species dependency on ISAM option results. The results of OP1, OP4, and OP5 can be explained in Section 2 in Lines 118–139. In Figure 7, BCON RNOx in OP1, OP4, and OP5 match the spatial distribution of total RNOx where hot spots are captured over high RNOx concentrations, indicating OP5 could switch more to OP4 over these locations. We expected OP2 and OP3 to produce comparable results to OSAT and BF because these two options were forced to assign more sources with tracked nitrate species in Table 1 as designed. It demonstrated that OP2 is more suitable for RNOx attributions, broadly, and also for those species that can quickly circulate. However, OP1, OP4, and OP5 can still be useful for other species, like the VOCs that we have presented in this paper or other primary pollutants. This is one of the reasons we added the ISAM flexibility for the user to select. We also added more lines in the conclusion to discuss ISAM's choice for RNOx in Lines 531–538, "*After assigning products to sources emitting nitrogen reactants, the OP2 option can predict results of $RNO_x$ attributions that are more comparable to OSAT and BF. It demonstrated that the OP2 works better for $RNO_x$ because it makes it easier to find the original source and lessens the effect of other sources when these species are cycling quickly through an integrated chemical reaction system. Unlike $O_3$ and $RNO_x$, the VOC contribution for the majority of source categories depends very little on the ISAM option. We expect that the user will use OP5 for $O_3$ and OP2 for $RNO_x$, but this is not a firm suggestion. In turn, we give the user this flexibility so that ISAM can be used for a wide range of purposes*."

**Lines 485-489. There are also significant differences between OSAT and OP5 for O3 apportionment to EGU, NONROAD, and ONROAD sources. These are sources for which it is important to estimate their O3 contributions accurately. Again, the authors need to describe in detail how the source apportionments are done in OP1 - OP5 and contrast those procedures with how the apportionment is done in OSAT so that the reader has some understanding of why these differences occur. Just stating that the procedures differ is not very helpful.**

We have expanded our explanations of each ISAM option in section 2 in response to the other concerns from the reviewers.

**Line 496. OILGAS appears to be about as large as AREA in contribution to VOC (Figures 5 and 8).**

They look similar in the map figures, but their scales are different. The design of these map columns is to compare each source apportionment method for each sector. We decided not to unify the scale for all sectors to better investigate the spatial distribution of contributions from each sector. In the supplement, we calculated their domain-wide averaged contributions (Table S2(a-b)); AREA typically contributes 3-4 ppb (> 7% of total) VOC, while OILGAS contributes less than 3 ppb (< 6% of total).

**Technical corrections**

First, we sincerely thank you for your carefulness and patience in checking citations and bibliographies. As they are automatically generated by software, sometimes they contain bugs or unexpected errors. We have updated them based on your comments.

**Line 38. Lefohn et al., 1998 citation is not in the reference list.**

Added.

Lefohn A. S., Shadwick D. S. and Ziman S. D., 1998. The Difficult Challenge of Attaining EPA's New Ozone Standard. Environmental Science & Technology. 32(11):276A-282A.

**Line 77. Sillman, 1996 citation is not in the reference list.**

Added.

Sillman, Sanford. "The use of NO y, H2O2, and HNO3 as indicators for ozone-NO x-hydrocarbon sensitivity in urban locations." Journal of Geophysical Research: Atmospheres 100, no. D7 (1995): 14175-14188.

**Line 90. 2016a should be 2016.**

Changed.

**Line 93. Baker and Kelly, 2014 and Baker and Woody, 2017 are not in the reference list.**

Removed the first one and updated the second.

Baker, K. R., M. C. Woody, G. S. Tonnesen, W. Hutzell, H. O. T. Pye, M. R. Beaver, G. Pouliot, and T. Pierce. "Contribution of regional-scale fire events to ozone and PM2. 5 air quality estimated by photochemical modeling approaches." Atmospheric Environment 140 (2016): 539-554.

**Line 121.  Pierce et al. 1999 is not in the reference list.**

Removed.

**Line 171.  Henderson et al., 2014 is not in the reference list.**

Added.

Henderson, B. H., F. Akhtar, H. O. T. Pye, S. L. Napelenok, and W. T. Hutzell. "A database and tool for boundary conditions for regional air quality modeling: description and evaluation." Geoscientific Model Development 7, no. 1 (2014): 339-360.

**Line 176.  Bash et al., 2016 is not in the reference list.**

Added.

Bash JO, Baker KR, Beaver MR, 2016. Evaluation of improved land use and canopy representation in BEIS v3. 61 with biogenic VOC measurements in California. Geoscientific Model Development 9, 2191.

**Line 360-361.  Burr and Zhang, 2011 is not in the reference list.  Jiminez and Baldano,2004 is Jiminez, 2004?**

Added the first but keep the second.

Burr, Michael J., and Yang Zhang. "Source apportionment of fine particulate matter over the Eastern US Part I: source sensitivity simulations using CMAQ with the Brute Force method." Atmospheric Pollution Research 2, no. 3 (2011): 300-317.

**Lines 580-581.  There are strings of symbols here that are unintelligible.**

Updated.

Kwok, R.H.F., Baker, K.R., Napelenok, S.L. and Tonnesen, G.S., 2015. Photochemical grid model implementation and application of VOC, NO x, and O 3 source apportionment. Geoscientific Model Development, 8(1), pp.99-114.

**Lines 650-653.  There are two U.S. EPA (2021) references.  These should be labeled 2021a and 2021b and cited as such.**

Updated.

**Line 660.  1967 or 1984?**

Cited as google scholar. 1984 is a newer version for 1967 version.

**The following publications are in the reference list but I did not find them cited in the text: Baker and Timin (2008); Oltmans et al. (1998); Sarwar et al. (2011)**

Removed.

**Response to reviewer 2**

https://gmd.copernicus.org/preprints/gmd-2022-273#RC2

The ISAM is a powerful tool for source apportionment of O3 and PM2.5 in CMAQ. The authors have updated ISAM with more attribution options and compared the results with different options to those using OSAT or the brute force method. Generally, the manuscript is well written, and the work is worthy of publication. There are a few questions that need to be addressed:

It is not quite clear in what cases or on what purpose is each of the option the best one? For example, is ISAM-OP2 more suitable for RNOx attribution? The authors could elaborate more.

We have added more sentences and discussions in conclusion section to elaborate it as below.

Lines 531-538 "*After assigning products to sources emitting nitrogen reactants, the OP2 option can predict results of RNOx attributions that are more comparable to OSAT and BF. It demonstrated that the OP2 works better for RNOx because it makes it easier to find the original source and lessens the effect of other sources when these species are cycling quickly through an integrated chemical reaction system. Unlike O3 and RNOx, the VOC contribution for the majority of source categories depends very little on the ISAM option. We expect that the user will use OP5 for O3 and OP2 for RNOx, but this is not a firm suggestion. In turn, we give the user this flexibility so that ISAM can be used for a wide range of purposes.*"

The CB6R3 and CB6R4 were used in CMAQ and CAMx, respectively. What are the impacts of using different chemical mechanisms?

Please see our response to the second major concern of the first reviewer.

Some mistakes in the manuscript. For example, lines 76-77: "when the ratio (PH2O2/PHNO3) is below 0.35, the formation is classified as NOx-limited…"; lines 199-200: "when the ratio of PH2O2/PHNO3 exceeds 0.35, the produced O3 is attributed to VOC emissions…"

We have corrected all similar errors.

**Response to chief editor**

**Dear authors,**

**this manuscript is definitely not of the type "methods of model assessment paper" as which it was submitted. It is either of type "development and technical paper" or a "model evaluation" paper.**

**For a "model evaluation paper", the new ISAM must have been per-reviewed published. This is, as far as I can see, not the case, as the only documentations which are cited in the manuscript are US EPA, 2021 (2 different documents / archives) and US EPA 2022, which all are not per-reviewed literature.**

**For a "development and technical paper" the details on the new scheme need to be provided within the article. Which seems to me to be not the case by now.**

**Anyhow, the detailed documentation of the new scheme cited in the article is US EPA 2021. This relates to https://www.epa.gov/air-emissions-modeling/2016-version-1-technical-support-document . This is a web site which could change its content any time and this is in any case not sufficient as documentation for a GMD publication. The content of this web site needs either to be archived permanently as well (in form of a document with DOI) or as supplement to this article or you have to include all the for ISAM relevant information into your article to meet the requirements of a "development and technical paper".**

**I will write to the copernicus office to change the type of the paper to "development and technical paper", please include the required documentation into your manuscript.**

**Furthermore, the modified code needs to be archived permanently. An "available on request statement" for your codes updates and scripts is not sufficient for publication in GMD**

**Best regards, Astrid Kerkweg (GMD Executive Editor)**

Thanks for your suggestions. We have modified the citations you mentioned in our paper. These citations are referred to the Zenodo archives that include the versions of CMAQ and ISAM we have used for this study. Relevant changes about ISAM updates and available code are updated in the paper as well.

We also updated our code availability in Line 554-559 *"The updated ISAM code used in this study has been permanently archived at https://doi.org/10.5281/zenodo.6266674 and has also been implemented in the newer version of CMAQ (v5.4). The CMAQ model documentation is available at https://github.com/USEPA/CMAQ and www.cmaq-model.org."*

**References**

1. Gillani, N. V., & Pleim, J. E. (1996). Sub-grid-scale features of anthropogenic emissions of NOx and VOC in the context of regional Eulerian models. *Atmospheric Environment*, *30*(12), 2043-2059.
2. Jacquemin, B. and Noilhan, J.: Sensitivity study and validation of a land surface parameterization using the HAPEX-MOBILHY data set, Boundary-Layer Meteorol, 52, 93–134, https://doi.org/10.1007/BF00123180, 1990.
3. Kwok, R.H.F., Baker, K.R., Napelenok, S.L. and Tonnesen, G.S., 2015. Photochemical grid model implementation and application of VOC, NO x, and O 3 source apportionment. Geoscientific Model Development, 8(1), pp.99-114.
4. Ramboll Environ. CAMx user guide v7.10. https://camx.com/Files/CAMxUsersGuide_v7.10.pdf
5. Sarwar, G., Gantt, B.; Schwede, D.; Foley, K.; Mathur, R.; Saiz-Lopez, A. Impact of enhanced ozone deposition and halogen chemistry on tropospheric ozone over the Northern Hemisphere, *Environmental Science & Technology*, 2015, **49**(15):9203-9211.
6. Sarwar, G.; Gantt, B.; Foley, K.; Fahey, K.; Spero T. L.; Kang, D., Mathur, Rohit M., Hosein F.; Xing, J.; Sherwen, T.; Saiz-Lopez, A., 2019: Influence of bromine and iodine chemistry on annual, seasonal, diurnal, and background ozone: CMAQ simulations over the Northern Hemisphere, *Atmospheric Environment*, 213, 395-404.
7. Sillman, Sanford. "The use of NO y, H2O2, and HNO3 as indicators for ozone-NO x-hydrocarbon sensitivity in urban locations." Journal of Geophysical Research: Atmospheres 100, no. D7 (1995): 14175-14188.
8. Emery, C., Liu, Z., Koo, B., Yarwood, G. 2016. Improved Halogen Chemistry for CAMx Modeling. Final report for Texas Commission on Environmental Quality WO 582-16-61842-13, May 2016, available at https://www.tceq.texas.gov/airquality/airmod/project/pj_report_pm.html (last accessed 13 December 2019).
9. Yienger, J. J. and Levy, H.: Empirical model of global soil-biogenic NOx emissions, J. Geophys. Res., 100, 11447, https://doi.org/10.1029/95JD00370, 1995.

---

## Referee Report (RR1)

One remaining issue needs to be addressed before the publication of this revised version of the manuscript: the explanation of why the bulk concentrations of O3, VOC etc. are found quite different between CMAQ and CMAx is not specific but more like a guess, and hence not convincing, though the concentration difference seems can explain the difference of apportionments between OSAT and ISAM.

I list some of the related such weak explanations as below:

Line 402-407 Given that the difference is present in the total concentration, this is unlikely caused by different source apportionment formulation between CMAQ and CAMx. As CAMx only gives pre-lumped VOC, it is challenging to compare individual VOC species between CMAQ and CAMx to explain this difference at current stage. Another possible reasons to cause it could be that models have different internal treatments for advection and diffusion, which can impact surface-level concentrations and indirectly impact chemical reactions.

Line 424-428: In Fig. 6, OSAT exhibits the same spatial 425 distribution of MDA8 O3 total concentrations as other CMAQ-based simulations (OP1, OP2, OP3, OP4, OP5, and CMAQ-BF), with the exception of OSAT's relatively high marine and offshore total concentrations (> 5 ppbv), which could be explained by the difference in planetary boundary layer dynamics or different marine chemistry configuration between the two parent models.

Line 452-454: As CMAQ and CAMx both use the same BEIS inventory data, the difference in total VOC concentrations may result from other differences between two models, like chemistry or deposition, accordingly, leading to higher biogenic sources in CAMx (BIO).

The actual confusion to the audience is that on one hand the authors claim that "all base meteorological and emissions inputs for CAMx were identical to those for CMAQ", and from the Table2, it shows that the inputs of ic and bc seem the same as well. The only known explicit difference is the halogen and DMS chemistry in the mechanisms, but as the authors stated that the resulted difference should be small over the land which has been demonstrated by other studies. On the other hand, significant differences in concentrations and hourly performances are found between the two models. For example, the Tables 5 and 6 (NMB and R of hourly NO2 and O3), and the following statement:

Line 359-361: In Fig. 3, CMAQ and CAMx predict 360 similar O3 concentrations during the day, but differences appear at night, with a maximum difference of 5 ppb. This disparity was discussed in Section 4.1 and can be mitigated by employing the MDA8 O3 metric.

Such differences in performances and in the simulated hourly concentrations, plus some unexpected differences in spatial patterns of the results, it seems there are differences existing in the meteorology used in the simulations.  Suggestions:

(1) Compare the simulated spatial fields of hourly concentrations for a couple of inert species, such as EC (or even CO if no other inert species can be found), especially look for differences during the night. The purpose is to check or demonstrate if the input meteorological fields are really identical between the two models' simulations.  If significantly different (most likely), then
(2) Compare the Kzz values between the two models. In CMAQ, KZMIN is a default option for setting Kzz cutoffs to limit Kzz values over different landuse during the night. In CAMx, there is no such runtime choice, but it provides a KVPATCH program to implement such fixes to the WRFCAMX outputs. Were KVPATCH outputs used as inputs to CAMx?

Other comments:

Table 2: It would be better to also list the model options in the table 2, such as advection, diffusion, and deposition schemes, chemistry solver, aero modules etc. In the addition, are the BC identical? It is only stated that the BC are from 12km simulations, but it doesn't say it is from the same CMAQ (or CAMx) 12km simulations or each from their own model's 12km simulations.

Line 64-65: Further, to separate the contributions and interactions of "n" sources, Stein and Alpert (1993) showed that BF would require two to the power of the number of sources (2n 65 ).

… require two to the power of the number of sources of simulations?

Line 263-265: As for OTHR, there is no suitable way to retain an appropriate chemical state of the troposphere after subtracting necessary emission categories, initial and boundary conditions from an original CMAQ simulation.

What is exactly OTHR? Does it mean the apportionments of IC, BC and all emissions categories don't add up to 100%? Why not?

---

## Author Response (AR2)

Dear Editor Jason Williams,

Thanks again for taking your time to organize another round of review. I am submitting the amended version of the manuscript as well as our response to the comments for this minor revision. My co-authors and I would like to thank additional reviewers for their valuable suggestions. I think we have addressed all the concerns accordingly, which has helped us make a much better article that we hope you will publish. In the pages attached, you will see details on how we addressed each reviewer's comment, as well as the revised language that followed.

We appreciate your time.

Sincerely,

Qian Shu

**Comparison of Ozone Formation Attribution Techniques in the Northeast United States**

Qian Shu[1], Sergey L. Napelenok[1], William T. Hutzell[1], Kirk R. Baker[1], Barron H. Henderson[1], Benjamin Murphy[1], Christian Hogrefe[1]

[1]U.S. Environmental Protection Agency, Research Triangle Park, NC, 27711, USA.

Correspondence to: Sergey L Napelenok (sergey.napelenok@epa.gov)

**Response to Anonymous Referee #3**

First, I would like to thank you for the helpful comments. Each comment is addressed accordingly.

**One remaining issue needs to be addressed before the publication of this revised version of the manuscript: the explanation of why the bulk concentrations of O3, VOC etc. are found quite different between CMAQ and CMAx is not specific but more like a guess, and hence not convincing, though the concentration difference seems can explain the difference of apportionments between OSAT and ISAM. I list some of the related such weak explanations as below:**

**Line 402-407 Given that the difference is present in the total concentration, this is unlikely caused by different source apportionment formulation between CMAQ and CAMx. As CAMx only gives pre-lumped VOC, it is challenging to compare individual VOC species between CMAQ and CAMx to explain this difference at current stage. Another possible reasons to cause it could be that models have different internal treatments for advection and diffusion, which can impact surface-level concentrations and indirectly impact chemical reactions.**

**Line 424-428: In Fig. 6, OSAT exhibits the same spatial 425 distribution of MDA8 O3 total concentrations as other CMAQ-based simulations (OP1, OP2, OP3, OP4, OP5, and CMAQ-BF), with the exception of OSAT's relatively high marine and offshore total concentrations (> 5 ppbv), which could be explained by the difference in planetary boundary layer dynamics or different marine chemistry configuration between the two parent models.**

**Line 452-454: As CMAQ and CAMx both use the same BEIS inventory data, the difference in total VOC concentrations may result from other differences between two models, like chemistry or deposition, accordingly, leading to higher biogenic sources in CAMx (BIO).**

**The actual confusion to the audience is that on one hand the authors claim that "all base meteorological and emissions inputs for CAMx were identical to those for CMAQ", and from the Table2, it shows that the inputs of ic and bc seem the same as well. The only known explicit difference is the halogen and DMS chemistry in the mechanisms, but as the authors stated that the resulted difference should be small over the land which has been demonstrated by other studies. On the other hand, significant differences in concentrations**

**and hourly performances are found between the two models. For example, the Tables 5 and 6 (NMB and R of hourly NO2 and O3), and the following statement:**
**Line 359-361: In Fig. 3, CMAQ and CAMx predict 360 similar O3 concentrations during the day, but differences appear at night, with a maximum difference of 5 ppb. This disparity was discussed in Section 4.1 and can be mitigated by employing the MDA8 O3 metric.**
**Such differences in performances and in the simulated hourly concentrations, plus some unexpected differences in spatial patterns of the results, it seems there are differences existing in the meteorology used in the simulations. Suggestions:**
**(1) Compare the simulated spatial fields of hourly concentrations for a couple of inert species, such as EC (or even CO if no other inert species can be found), especially look for differences during the night. The purpose is to check or demonstrate if the input meteorological fields are really identical between the two models' simulations. If significantly different (most likely), then**
**(2) Compare the Kzz values between the two models. In CMAQ, KZMIN is a default option for setting Kzz cutoffs to limit Kzz values over different landuse during the night. In CAMx, there is no such runtime choice, but it provides a KVPATCH program to implement such fixes to the WRFCAMX outputs. Were KVPATCH outputs used as inputs to CAMx?**

First, it is impossible for the models to have the same bulk prediction or tagged contribution; thus, we want to be clear that identical results are not what we anticipate or want. The main goals of this paper are to 1) document how we recently made ISAM options for secondary species, especially ozone, and 2) evaluate how these ISAM options work. We used the brute force approach and CAMx-OSAT as references because there isn't a perfect reference for assessing our methods. There are a lot of uncertainties when comparing CMAQ-ISAM and CAMx-OSAT, including the ones we mentioned in the manuscript (Lines 285-290) and the ones the reviewer brought up in the comments. We don't think the two models will be the same when it comes to bulk prediction or tagged contribution because they are quite different in many ways, both internally and in their source apportionment methods. To show that our ISAM model is reasonably designed, we anticipate that the results of ISAM should approach OSAT or be reasonable in common sense. Therefore, in addition to OSAT, we also add brute force as another reference since our brute force simulations are running under the CMAQ parent model, in which case we do not need to worry about parent differences. However, there are still imperfections in the comparison between CMAQ-ISAM and CMAQ-BF for secondary species because CMAQ-BF also has its own limitations for nonlinear chemistry (Lines 56–68). To minimize the impact of uncertainties from parent models, we constrained all controllable elements by (1) providing identical inputs as stated in Table 2, (2) providing the same IC and BC, and (3) using the same or at least comparable model processes.one exception is the chemical mechanism because there is not the same option for the model versions we have tested in this study. Even if these variables are tightly controlled, there are structural and parametric differences between the models. The structural differences include different algorithms applied for chemistry and physical processes, which includes different numerical approximations during implementation of identical algorithms. The parametric differences include rate coefficients (JPL vs IUPAC vs and adjustments), physical properties (e.g., surrogate properties for lumped species), and different criteria for computational time-step adjustment. These differences manifest differently by species and cannot be universally explained without performing extensive additional research on each

species, property, and process representation. There are already several studies on model comparisons that show differences with the same inputs (Tesche et al., 2005; Koo et al., 2014; Dolwich et al., 2015; 2014; Shu et al., 2017). The explanation that the reviewer requests is outside the scope of this paper. There are several approaches that could be applied in the future to isolate the difference more perfectly. Future research should either implement source apportionment approaches, both ISAM and OSAT, to the same parent, CAMx or CMAQ, or use process analysis to quantify each process for each species, such as ozone and individual VOCs to comprehend the distinction between two parent models. It requires a significant amount of extra work and is beyond the scope of this work. As a result, we are unable to conclusively explain the bulk difference.

We plotted the EC for CMAQ and CAMx for the two-day case simulations as you asked. EC is an inert species in both models. The outcomes of the two models are anticipated to be somewhat similar but not the same. CMAQ EC is overall higher than CAMx most of the time in the two-day case simulation. This is an expected outcome since we have updated particle dry deposition for CMAQ v5.3 which tends to slow down particle deposition, leading to higher EC concentrations in the surface atmosphere (Shu et al., 2022). A similar EC analysis can also be found in Shu et al. (2017). This is also a good example of how other process uncertainties influence the concentration results, even for chemically inert species when identical inputs are given. Again, we can't draw the conclusion that this EC difference is only caused by deposition without performing process analysis to quantify the influence from other processes. To be clear, the dry deposition update in CMAQv5.3 only affects particles and should not influence gas dry deposition for ozone and VOCs in this study. However, differences in the representation of gas dry deposition exist between the two models (Shu et al., 2017), and this is just one example of how process representations can and often do differ between the two models. In this case, it takes extra efforts in process analysis for more complicated chemical species like ozone and VOC to understand what causes model differences. We do agree it is valuable to understand it and provide a good explanation for that, but this is beyond the scope of the current study. We think more work can be done in the future, as we previously discussed both in the manuscript and in our response here. We are not going to add this EC analysis to either our main manuscript or supplement but keep it here as a response to your comment.

[Figure]

**Other comments:**

**Table 2: It would be better to also list the model options in the table 2, such as advection, diffusion, and deposition schemes, chemistry solver, aero modules etc. In the addition, are the BC identical? It is only stated that the BC are from 12km simulations, but it doesn't say it is from the same CMAQ (or CAMx) 12km simulations or each from their own model's 12km simulations.**

We have added more configuration information in the Table 2 as below. BC and IC are the same. To be clearer, we added the sentence "Identical ICs and BCs were applied to the two models" In Lines 219-220.

Table 2. CMAQ and CAMx model configurations

| Model option | CMAQ | CAMx |
|---|---|---|
| Model version | Version 5.3.2 | Version 7.10 |
| Horizontal resolution | 4 km | 4 km |
| Vertical layers | 35 | 35 |
| Meteorology | WRF3.8 | WRF3.8 |
| Anthropogenic emissions | 2016 NEI version 1[a] | 2016 NEI version 1[b] |
| Biogenic emissions | BEIS[c] | BEIS[c] |
| BC/IC | 12km US CONUS | 12km US CONUS |
| Gas phase chemistry | CB6R3 | CB6R4 |
| Chemistry solver | EBI | EBI |
| Aerosol dynamics and chemistry | AERO7/ISORROPIA | SOAP/ISORROPIA |
| Horizontal advection | PPM | PPM |
| Vertical advection | PPM | Emery et al. (2011)[d] |
| Horizontal diffusion | Implicit[e] | Explicit simultaneous 2-D solver |

| | | |
|---|---|---|
| Vertical diffusion | ACM2[f] | Based on ACM2[g] |
| Gas deposition | Pleim and Ran (2011) | Zhang et al. (2003) |
| Particle deposition | Shu et al. (2022) | Zhang et al. (2001) |
| Source apportionment | ISAM | OSAT3 |

[a]EGU were based on continuous emissions monitoring data from 2018 where available. Onroad emissions were projected to 2018.
[b]CAMx EGU and Onroad were identically processed as CMAQ.
[c]BELD v4.1 vegetation data for biogenic emissions, BEIS version is 3.61.
[d]Backward-Euler (time) hybrid centered/upstream (space) solver.
[e]Horizontal diffusion fluxes for transported pollutants were parameterized using eddy diffusion theory. The horizontal diffusivity coefficients were formulated using the approach of Smagorinsky (1963).
[f]KZMIN was turned on in CMAQ as default.
[g]Vertical diffusivity coefficients were calculated with Yonsei University (YSU) bulk boundary layer scheme (Hong et al., 2006) and were adjusted with the KVPATCH which is comparable to the KZMIN approach in CMAQ.

**Line 64-65: Further, to separate the contributions and interactions of "n" sources, Stein and Alpert (1993) showed that BF would require two to the power of the number of sources (2n 65 ).**
**… require two to the power of the number of sources of simulations?**

We have changed it to 'the number of sources of simulations'

**Line 263-265: As for OTHR, there is no suitable way to retain an appropriate chemical state of the troposphere after subtracting necessary emission categories, initial and boundary conditions from an original CMAQ simulation.**
**What is exactly OTHR? Does it mean the apportionments of IC, BC and all emissions categories don't add up to 100%? Why not?**

In Lines 258–260. "OTHR is used for all remaining untagged emission categories. In this study, all emissions sectors were tracked as previously mentioned above for OSAT and ISAM." When it comes to OTHR, the user can define how many sectors to track in ISAM. For example, when there are a total of ten emission streams but only five of them are tracked in ISAM, the remaining five emission streams will be defined as OTHR. We have also added the previous sentence in Lines 258–260 to make it clearer for readers. Which is to say, the apportionments of IC, BC, user-defined emission tags, and remaining untagged emission categories should add up to 100%. It may not be perfectly summed up to 100% of the bulk concentrations, but it should match the bulk concentrations very closely.

References

1. Dolwick, P., Akhtar, F., Baker, K. R., Possiel, N., Simon, H., & Tonnesen, G. (2015). Comparison of background ozone estimates over the western United States based on two separate model methodologies. Atmospheric Environment, 109, 282-296.

2. Emery, C., E. Tai, G. Yarwood, R. Morris. 2011. Investigation into approaches to reduce excessive vertical transport over complex terrain in a regional photochemical grid model. Atmos. Environ., 45, 7341-7351, doi:10.1016/j.atmosenv.2011.07.052.

3. Hong, S. Y., Noh, Y., & Dudhia, J. (2006). A new vertical diffusion package with an explicit treatment of entrainment processes. Monthly weather review, 134(9), 2318-2341.

4. Koo, B., Knipping, E., & Yarwood, G. (2014). 1.5-Dimensional volatility basis set approach for modeling organic aerosol in CAMx and CMAQ. Atmospheric Environment, 95, 158-164.

5. Pleim, J., Ran, L., 2011. Surface flux modeling for air quality applications. Atmosphere 2, 271–302. http://dx.doi.org/10.3390/atmos2030271.

6. Shu, Q., Koo, B., Yarwood, G., and Henderson, B. H.: Strong influence of deposition and vertical mixing on secondary organic aerosol concentrations in CMAQ and CAMx, Atmospheric Environment, 171, 317–329, https://doi.org/10.1016/j.atmosenv.2017.10.035, 2017.

7. Shu, Q., Murphy, B., Schwede, D., Henderson, B. H., Pye, H. O., Appel, K. W., ... & Perlinger, J. A. (2022). Improving the particle dry deposition scheme in the CMAQ photochemical modeling system. Atmospheric Environment, 289, 119343.

8. Smagorinsky, J. (1963). General circulation experiments with the primitive equations. Mon. Wea. Rev., 91/3, 99-164.

9. Tesche, T. W., Morris, R., Tonnesen, G., McNally, D., Boylan, J., & Brewer, P. (2006). CMAQ/CAMx annual 2002 performance evaluation over the eastern US. Atmospheric Environment, 40(26), 4906-4919.

10. Zhang, L., S. Gong, J. Padro, L. Barrie. 2001. A size-segregated particle dry deposition scheme for an atmospheric aerosol module. Atmos. Environ., 35, 549-560

11. Zhang, L., Brook, J.R., Vet, R., 2003. A revised parameterization for gaseous dry deposition in air-quality models. Atmos. Chem. Phys. 3, 2067–2082. http://dx.doi.org/ 10.5194/acp-3-2067-2003.